# A Matrix Chernoff Bound for Markov Chains and Its Application to Co-occurrence Matrices

**Jiezhong Qiu**
Tsinghua University
qiujz16@mails.tsinghua.edu.cn

**Chi Wang**
Microsoft Research, Redmond
wang.chi@microsoft.com

**Ben Liao**
Tencent Quantum Lab
bliao@tencent.com

**Richard Peng**
Georgia Tech
rpeng@cc.gatech.edu

**Jie Tang**
Tsinghua University
jietang@tsinghua.edu.cn

## Abstract

We prove a Chernoff-type bound for sums of matrix-valued random variables sampled via a regular (aperiodic and irreducible) finite Markov chain. Specially, consider a random walk on a regular Markov chain and a Hermitian matrix-valued function on its state space. Our result gives exponentially decreasing bounds on the tail distributions of the extreme eigenvalues of the sample mean matrix. Our proof is based on the matrix expander (regular undirected graph) Chernoff bound [Garg et al. STOC '18] and scalar Chernoff-Hoeffding bounds for Markov chains [Chung et al. STACS '12].

Our matrix Chernoff bound for Markov chains can be applied to analyze the behavior of co-occurrence statistics for sequential data, which have been common and important data signals in machine learning. We show that given a regular Markov chain with $n$ states and mixing time $\tau$, we need a trajectory of length $O(\tau(\log n + \log \tau)/\epsilon^2)$ to achieve an estimator of the co-occurrence matrix with error bound $\epsilon$. We conduct several experiments and the experimental results are consistent with the exponentially fast convergence rate from theoretical analysis. Our result gives the first bound on the convergence rate of the co-occurrence matrix and the first sample complexity analysis in graph representation learning.

## 1 Introduction

Chernoff bound [5], which gives exponentially decreasing bounds on tail distributions of sums of independent scalar-valued random variables, is one of the most basic and versatile tools in theoretical computer science, with countless applications to practical problems [21, 35]. There are two notable limitations when applying Chernoff bound in analyzing sample complexity in real-world machine learning problems. First, in many cases the random variables have dependence, e.g., Markov dependence [20] in MCMC [18] and online learning [48]. Second, applications are often concerned with the concentration behavior of quantities beyond scalar-valued random variables, e.g., random features in kernel machines [40] and co-occurrence statistics which are random matrices [38, 39].

Existing research has attempted to extend the original Chernoff bound in one of these two limitations [19, 11, 27, 24, 53, 14, 6, 41, 52, 42, 1, 50]. Wigderson and Xiao [53] conjectured that Chernoff bounds can be generalized to both matrix-valued random variables and Markov dependence, while restricting the Markov dependence to be *a regular undirected graph*. It was recently proved by Garg et al. [10], based on a new multi-matrix extension of the Golden-Thompson inequality [45]. However, the regular undirected graph is a special case of Markov chains which are reversible and have a uniform stationary distribution, and does not apply to practical problems such as random walk on

generic graphs. It is an open question for the Chernoff bound of matrix-valued random matrices with more generic Markov dependence.

In this work, we establish large deviation bounds for the tail probabilities of the extreme eigenvalues of sums of random matrices sampled via a regular Markov chain[1] starting from an arbitrary distribution (not necessarily the stationary distribution), which significantly improves the result of Garg et al. [10]. More formally, we prove the following theorem:

**Theorem 1** (Markov Chain Matrix Chernoff Bound). *Let $P$ be a regular Markov chain with state space $[N]$, stationary distribution $\pi$ and spectral expansion $\lambda$. Let $f : [N] \to \mathbb{C}^{d \times d}$ be a function such that (1) $\forall v \in [N]$, $f(v)$ is Hermitian and $\|f(v)\|_2 \leq 1$; (2) $\sum_{v \in [N]} \pi_v f(v) = 0$. Let $(v_1, \cdots, v_k)$ denote a $k$-step random walk on $P$ starting from a distribution $\phi$. Given $\epsilon \in (0, 1)$,*

$$\mathbb{P}\left[\lambda_{\max}\left(\frac{1}{k}\sum_{j=1}^{k} f(v_j)\right) \geq \epsilon\right] \leq 4 \|\phi\|_{\pi} \, d^2 \exp\left(-(\epsilon^2 (1-\lambda)k/72)\right)$$

$$\mathbb{P}\left[\lambda_{\min}\left(\frac{1}{k}\sum_{j=1}^{k} f(v_j)\right) \leq -\epsilon\right] \leq 4 \|\phi\|_{\pi} \, d^2 \exp\left(-(\epsilon^2 (1-\lambda)k/72)\right).$$

In the above theorem, $\|\cdot\|_{\pi}$ is the $\pi$-norm (which we define formally later in Section 2) measuring the distance between the initial distribution $\phi$ and the stationary distribution $\pi$. Our strategy is to incorporate the concentration of matrix-valued functions from [10] into the study of general Markov chains from [6], which was originally for scalars.

## 1.1 Applications to Co-occurrence Matrices of Markov Chains

---
**Algorithm 1:** The Co-occurrence Matrix.

---
1 **Input** sequence $(v_1, \cdots, v_L)$; window size $T$;
2 **Output** co-occurrence matrix $C$;
3 $C \leftarrow \mathbf{0}_{n \times n}$; ;          /* $v_i \in [n], i \in [L]$ */
4 **for** $i = 1, 2, \ldots, L - T$ **do**
5     **for** $r = 1, \ldots, T$ **do**
6         $C_{v_i, v_{i+r}} \leftarrow C_{v_i, v_{i+r}} + 1/T$;
7         $C_{v_{i+r}, v_i} \leftarrow C_{v_{i+r}, v_i} + 1/T$;
8 $C \leftarrow \frac{1}{2(L-T)} C$;
9 **Return** $C$;

---

The co-occurrence statistics have recently emerged as common and important data signals in machine learning, providing rich correlation and clustering information about the underlying object space, such as the word co-occurrence in natural language processing [32–34, 26, 37], vertex co-occurrence in graph learning [38, 46, 12, 13, 7, 39], item co-occurrence in recommendation system [44, 28, 2, 51, 29], action co-occurrence in reinforcement learning [49], and emission co-occurrence of hidden Markov models [23, 17, 30]. Given a sequence of objects $(v_1, \cdots, v_L)$, the co-occurrence statistics are computed by moving a sliding window of fixed size $T$ over the sequence and recording the frequency of objects' co-occurrence within the sliding window. A pseudocode of the above procedure is listed in Algorithm 1, which produces an $n$ by $n$ co-occurrence matrix where $n$ is the size of the object space.

A common assumption when building such co-occurrence matrices is that the sequential data are long enough to provide an accurate estimation. For instance, Mikolov et al. [33] use a news article dataset with one billion words in their Skip-gram model; Tennenholtz and Mannor [49] train their Act2vec model with action sequences from over a million StarCraft II game replays, which are equivalent to 100 years of consecutive gameplay; Perozzi et al. [38] samples large amounts of random walk sequences from graphs to capture the vertex co-occurrence. A recent work by Qiu et al. [39] studies the convergence of co-occurrence matrices of random walk on undirected graphs in the limit (i.e., when the length of random walk goes to infinity), but left the convergence rate an open problem. It remains unknown whether the co-occurrence statistics are sample efficient and how efficient they are.

In this paper, we study the situation where the sequential data are sampled from a regular finite Markov chain (i.e., an aperiodic and irreducible finite Markov chain), and derive bounds on the sample efficiency of co-occurrence matrix estimation, specifically on the length of the trajectory needed in the sampling algorithm shown in Algorithm 1. To give a formal statement, we first translate

Algorithm 1 to linear algebra language. Given a trajectory $(v_1, \cdots, v_L)$ from state space $[n]$ and step weight coefficients $(\alpha_1, \cdots, \alpha_T)$, the co-occurrence matrix is defined to be

$$\boldsymbol{C} \triangleq \frac{1}{L-T} \sum_{i=1}^{L-T} \boldsymbol{C}_i, \text{where } \boldsymbol{C}_i \triangleq \sum_{r=1}^{T} \frac{\alpha_r}{2} \left( \boldsymbol{e}_{v_i} \boldsymbol{e}_{v_{i+r}}^{\top} + \boldsymbol{e}_{v_{i+r}} \boldsymbol{e}_{v_i}^{\top} \right).$$

Here $\boldsymbol{C}_i$ accounts for the co-occurrence within sliding window $(v_i, \cdots, v_{i+T})$, and $\boldsymbol{e}_{v_i}$ is a length-$n$ vector with a one in its $v_i$-th entry and zeros elsewhere. Thus $\boldsymbol{e}_{v_i} \boldsymbol{e}_{v_{i+r}}^{\top}$ is a $n$ by $n$ matrix with its $(v_i, v_{i+r})$-th entry to be one and other entries to be zero, which records the co-occurrence of $v_i$ and $v_{i+r}$. Note that Algorithm 1 is a special case when step weight coefficients are uniform, i.e., $\alpha_r = 1/T, r \in [T]$, and the co-occurrence statistics in all the applications mentioned above can be formalized in this way. When trajectory $(v_1, \cdots, v_L)$ is a random walk from a regular Markov chain $\boldsymbol{P}$ with stationary distribution $\boldsymbol{\pi}$, the asymptotic expectation of the co-occurrence matrix within sliding window $(v_i, \cdots, v_{i+L})$ is

$$\mathbb{AE}[\boldsymbol{C}_i] \triangleq \lim_{i \to \infty} \mathbb{E}(\boldsymbol{C}_i) = \sum_{r=1}^{T} \frac{\alpha_r}{2} \left( \boldsymbol{\Pi} \boldsymbol{P}^r + (\boldsymbol{\Pi} \boldsymbol{P}^r)^{\top} \right),$$

where $\boldsymbol{\Pi} \triangleq \text{diag}(\boldsymbol{\pi})$. Thus the asymptotic expectation of the co-occurrence matrix is

$$\mathbb{AE}[\boldsymbol{C}] \triangleq \lim_{L \to \infty} \mathbb{E}[\boldsymbol{C}] = \lim_{L \to \infty} \frac{1}{L-T} \sum_{i=1}^{L-T} \mathbb{E}(\boldsymbol{C}_i) = \sum_{r=1}^{T} \frac{\alpha_r}{2} \left( \boldsymbol{\Pi} \boldsymbol{P}^r + (\boldsymbol{\Pi} \boldsymbol{P}^r)^{\top} \right). \tag{1}$$

Our main result regarding the estimation of the co-occurrence matrix is the following convergence bound related to the length of the walk sampled.

**Theorem 2** (Convergence Rate of Co-occurrence Matrices). *Let $\boldsymbol{P}$ be a regular Markov chain with state space $[n]$, stationary distribution $\boldsymbol{\pi}$ and mixing time $\tau$. Let $(v_1, \cdots, v_L)$ denote a L-step random walk on $\boldsymbol{P}$ starting from a distribution $\phi$ on $[n]$. Given step weight coefficients $(\alpha_1, \cdots, \alpha_T)$ s.t. $\sum_{r=1}^{T} |\alpha_r| = 1$, and $\epsilon \in (0, 1)$, the probability that the co-occurrence matrix $\boldsymbol{C}$ deviates from its asymptotic expectation $\mathbb{AE}[\boldsymbol{C}]$ (in 2-norm) is bounded by:*

$$\mathbb{P}\left[ \|\boldsymbol{C} - \mathbb{AE}[\boldsymbol{C}]\|_2 \geq \epsilon \right] \leq 2 \left(\tau + T\right) \|\phi\|_{\boldsymbol{\pi}} n^2 \exp\left( -\frac{\epsilon^2 (L-T)}{576 \left(\tau + T\right)} \right).$$

*Specially, there exists a trajectory length $L = O\left((\tau + T)(\log n + \log(\tau + T))/\epsilon^2 + T\right)$ such that $\mathbb{P}\left[ \|\boldsymbol{C} - \mathbb{AE}[\boldsymbol{C}]\|_2 \geq \epsilon \right] \leq \frac{1}{n^{O(1)}}$. Assuming $T = O(1)$ gives $L = O\left(\tau(\log n + \log \tau)/\epsilon^2\right)$.*

Our result in Theorem 2 gives the first sample complexity analysis for many graph representation learning algorithms. Given a graph, these algorithms aim to learn a function from the vertices to a low dimensional vector space. Most of them (e.g., DeepWalk [38], node2vec [12], metapath2vec [7], GraphSAGE [13]) consist of two steps. The first step is to draw random sequences from a stochastic process defined on the graph and then count co-occurrence statistics from the sampled sequences, where the stochastic process is usually defined to be first-order or higher-order random walk on the graph. The second step is to train a model to fit the co-occurrence statistics. For example, DeepWalk can be viewed as factorizing a point-wise mutual information matrix [26, 39] which is a transformation of the co-occurrence matrix; GraphSAGE fits the co-occurrence statistics with a graph neural network [22]. The common assumption is that there are enough samples so that the co-occurrence statistics are accurately estimated. We are the first work to study the sample complexity of the aforementioned algorithms. Theorem 2 implies that these algorithms need $O(\tau(\log n + \log \tau)/\epsilon^2)$ samples to achieve a good estimator of the co-occurrence matrix.

**Previous work** Hsu et al. [16, 15] study a similar problem. They leverage the co-occurrence matrix with $T = 1$ to estimate the mixing time in reversible Markov chains from a single trajectory. Their main technique is a blocking technique [55] which is in parallel with the Markov chain matrix Chernoff-bound used in this work. Our work is also related to the research about random-walk matrix polynomial sparsification when the Markov chain $\boldsymbol{P}$ is a random walk on an undirected graph. In this case, we can rewrite $\boldsymbol{P} = \boldsymbol{D}^{-1} \boldsymbol{A}$ where $\boldsymbol{D}$ and $\boldsymbol{A}$ is the degree matrix and adjacency matrix of an undirected graph with $n$ vertices and $m$ edges, and the expected co-occurrence matrix in Equation 1 can be simplified as $\mathbb{AE}[\boldsymbol{C}] = \frac{1}{\text{vol}(G)} \sum_{r=1}^{T} \alpha_r \boldsymbol{D}(\boldsymbol{D}^{-1} \boldsymbol{A})^r$,[2] which is known as *random-walk*

*matrix polynomials* [3, 4]. Cheng et al. [4] propose an algorithm which needs $O(T^2 m \log n / \epsilon^2)$ steps of random walk to construct an $\epsilon$-approximator for the random-walk matrix polynomials. Our bound in Theorem 2 is stronger than the bound proposed by Cheng et al. [4] when the Markov chain $\boldsymbol{P}$ mixes fast. Moreover, Cheng et al. [4] require $\alpha_r$ to be non-negative, while our bound can handle negative step weight coefficients.

**Organization** The rest of the paper is organized as follows. In Section 2 we provide preliminaries, followed by the proof of matrix Chernoff bound in Section 3 and the proof of convergence rate of co-occurrence matrices in Section 4. In Section 5, we conduct experiments on both synthetic and real-world datasets. Finally, we conclude this work in Section 6.

## 2 Preliminaries

In this paper, we denote $\boldsymbol{P}$ to be a finite Markov chain on $n$ states. $\boldsymbol{P}$ could refer to either the chain itself or the corresponding transition probability matrix — an $n$ by $n$ matrix such that its entry $\boldsymbol{P}_{ij}$ indicates the probability that state $i$ moves to state $j$. A Markov chain is called an ergodic Markov chain if it is possible to eventually get from every state to every other state with positive probability. A Markov chain is regular if some power of its transition matrix has all strictly positive entries. A regular Markov chain must be an ergodic Markov chain, but not vice versa. An ergodic Markov chain has unique stationary distribution, i,e., there exists a unique probability vector $\boldsymbol{\pi}$ such that $\boldsymbol{\pi}^\top = \boldsymbol{\pi}^\top \boldsymbol{P}$. For convenience, we denote $\boldsymbol{\Pi} \triangleq \mathrm{diag}(\boldsymbol{\pi})$.

The time that a regular Markov chain[3] needs to be "close" to its stationary distribution is called *mixing time*. Let $\boldsymbol{x}$ and $\boldsymbol{y}$ be two probability vectors. The *total variation distance* between them is $\|\boldsymbol{x} - \boldsymbol{y}\|_{TV} \triangleq \frac{1}{2} \|\boldsymbol{x} - \boldsymbol{y}\|_1$. For $\delta > 0$, the $\delta$-mixing time of regular Markov chain $\boldsymbol{P}$ is $\tau(\boldsymbol{P}) \triangleq \min \left\{ t : \max_{\boldsymbol{x}} \left\| (\boldsymbol{x}^\top \boldsymbol{P}^t)^\top - \boldsymbol{\pi} \right\|_{TV} \leq \delta \right\}$, where $\boldsymbol{x}$ is an arbitrary probability vector.

The stationary distribution $\boldsymbol{\pi}$ also defines a inner product space where the inner product (under $\boldsymbol{\pi}$-kernel) is defined as $\langle \boldsymbol{x}, \boldsymbol{y} \rangle_{\boldsymbol{\pi}} \triangleq \boldsymbol{y}^* \boldsymbol{\Pi}^{-1} \boldsymbol{x}$ for $\forall \boldsymbol{x}, \boldsymbol{y} \in \mathbb{C}^N$, where $\boldsymbol{y}^*$ is the conjugate transpose of $\boldsymbol{y}$. A naturally defined norm based on the above inner product is $\|\boldsymbol{x}\|_{\boldsymbol{\pi}} \triangleq \sqrt{\langle \boldsymbol{x}, \boldsymbol{x} \rangle_{\boldsymbol{\pi}}}$. Then we can define the *spectral expansion* $\lambda(\boldsymbol{P})$ of a Markov chain $\boldsymbol{P}$ [31, 9, 6] as $\lambda(\boldsymbol{P}) \triangleq \max_{\langle \boldsymbol{x}, \boldsymbol{\pi} \rangle_{\boldsymbol{\pi}} = 0, \boldsymbol{x} \neq 0} \frac{\|(\boldsymbol{x}^* \boldsymbol{P})^*\|_{\boldsymbol{\pi}}}{\|\boldsymbol{x}\|_{\boldsymbol{\pi}}}$. The spectral expansion $\lambda(\boldsymbol{P})$ is known to be a measure of mixing time of a Markov chain. The smaller $\lambda(\boldsymbol{P})$ is, the faster a Markov chain converges to its stationary distribution [54]. If $\boldsymbol{P}$ is reversible, $\lambda(\boldsymbol{P})$ is simply the second largest absolute eigenvalue of $\boldsymbol{P}$ (the largest is always 1). The irreversible case is more complicated, since $\boldsymbol{P}$ may have complex eigenvalues. In this case, $\lambda(\boldsymbol{P})$ is actually the square root of the second largest absolute eigenvalue of the *multiplicative reversiblization* of $\boldsymbol{P}$ [9]. When $\boldsymbol{P}$ is clear from the context, we will simply write $\tau$ and $\lambda$ for $\tau(\boldsymbol{P})$ and $\lambda(\boldsymbol{P})$, respectively. We shall also refer $1 - \lambda(\boldsymbol{P})$ as the *spectral gap* of $\boldsymbol{P}$.

## 3 Matrix Chernoff Bounds for Markov Chains

This section provides a brief overview of our proof of Markov chain Martrix Chernoff bounds. We start from a simpler version which only consider real-valued symmetric matrices, as stated in Theorem 3 below. Then we extend it to complex-valued Hermitian matrices, as stated in in Theorem 1.

**Theorem 3** (A Real-Valued Version of Theorem 1). *Let $\boldsymbol{P}$ be a regular Markov chain with state space $[N]$, stationary distribution $\boldsymbol{\pi}$ and spectral expansion $\lambda$. Let $f : [N] \to \mathbb{R}^{d \times d}$ be a function such that (1) $\forall v \in [N]$, $f(v)$ is symmetric and $\|f(v)\|_2 \leq 1$; (2) $\sum_{v \in [N]} \pi_v f(v) = 0$. Let $(v_1, \cdots, v_k)$ denote a $k$-step random walk on $\boldsymbol{P}$ starting from a distribution $\phi$ on $[N]$. Then given $\epsilon \in (0, 1)$,*

$$\mathbb{P}\left[ \lambda_{\max}\left( \frac{1}{k} \sum_{j=1}^{k} f(v_j) \right) \geq \epsilon \right] \leq \|\phi\|_{\boldsymbol{\pi}} \, d^2 \exp\left( -(\epsilon^2 (1 - \lambda) k / 72) \right)$$

$$\mathbb{P}\left[ \lambda_{\min}\left( \frac{1}{k} \sum_{j=1}^{k} f(v_j) \right) \leq -\epsilon \right] \leq \|\phi\|_{\boldsymbol{\pi}} \, d^2 \exp\left( -(\epsilon^2 (1 - \lambda) k / 72) \right).$$

Due to space constraints, we defer the full proof to Section B in the supplementary material and instead present a sketch here. By symmetry, we only discuss on bounding $\lambda_{\max}$ here. Using the exponential method, the probability in Theorem 3 can be upper bounded for any $t > 0$ by:

$$\mathbb{P}\left[\lambda_{\max}\left(\frac{1}{k}\sum_{j=1}^{k}f(v_j)\right) \geq \epsilon\right] \leq \mathbb{P}\left[\operatorname{Tr}\left[\exp\left(t\sum_{j=1}^{k}f(v_j)\right)\right] \geq \exp\left(tk\epsilon\right)\right] \leq \frac{\mathbb{E}\left[\operatorname{Tr}\left[\exp\left(t\sum_{j=1}^{k}f(v_j)\right)\right]\right]}{\exp\left(tk\epsilon\right)},$$

where the first inequality follows by the tail bounds for eigenvalues (See Proposition 3.2.1 in Tropp [50]) which controls the tail probabilities of the extreme eigenvalues of a random matrix by producing a bound for the trace of the matrix moment generating function, and the second inequality follows by Markov's inequality. The RHS of the above equation is the expected trace of the exponential of a sum of matrices (i.e., $tf(v_j)$'s). When $f$ is a scalar-valued function, we can easily write exponential of a sum to be product of exponentials (since $\exp(a+b) = \exp(a)\exp(b)$ for scalars). However, this is not true for matrices. To bound the expectation term, we invoke the following multi-matrix Golden-Thompson inequality from [10], by letting $\boldsymbol{H}_j = tf(v_j), j \in [k]$.

**Theorem 4** (Multi-matrix Golden-Thompson Inequality, Theorem 1.5 in [10]). *Let $\boldsymbol{H}_1, \cdots \boldsymbol{H}_k$ be $k$ Hermitian matrices, then for some probability distribution $\mu$ on $[-\frac{\pi}{2}, \frac{\pi}{2}]$.*

$$\log\left(\operatorname{Tr}\left[\exp\left(\sum_{j=1}^{k}\boldsymbol{H}_j\right)\right]\right) \leq \frac{4}{\pi}\int_{-\frac{\pi}{2}}^{\frac{\pi}{2}}\log\left(\operatorname{Tr}\left[\prod_{j=1}^{k}\exp\left(\frac{e^{\mathrm{i}\phi}}{2}\boldsymbol{H}_j\right)\prod_{j=k}^{1}\exp\left(\frac{e^{-\mathrm{i}\phi}}{2}\boldsymbol{H}_j\right)\right]\right)d\mu(\phi).$$

The key point of this theorem is to relate the exponential of a sum of matrices to a product of matrix exponentials and their adjoints, whose trace can be further bounded via the following lemma by letting $e^{\mathrm{i}\phi} = \gamma + \mathrm{i}b$.

**Lemma 1** (Analogous to Lemma 4.3 in [10]). *Let $\boldsymbol{P}$ be a regular Markov chain with state space $[N]$ with spectral expansion $\lambda$. Let $f$ be a function $f : [N] \to \mathbb{R}^{d \times d}$ such that (1) $\sum_{v \in [N]} \pi_v f(v) = 0$; (2) $\|f(v)\|_2 \leq 1$ and $f(v)$ is symmetric, $v \in [N]$. Let $(v_1, \cdots, v_k)$ denote a $k$-step random walk on $\boldsymbol{P}$ starting from a distribution $\phi$ on $[N]$. Then for any $t > 0, \gamma \geq 0, b > 0$ such that $t^2(\gamma^2 + b^2) \leq 1$ and $t\sqrt{\gamma^2 + b^2} \leq \frac{1-\lambda}{4\lambda}$, we have*

$$\mathbb{E}\left[\operatorname{Tr}\left[\prod_{j=1}^{k}\exp\left(\frac{tf(v_j)(\gamma+\mathrm{i}b)}{2}\right)\prod_{j=k}^{1}\exp\left(\frac{tf(v_j)(\gamma-\mathrm{i}b)}{2}\right)\right]\right] \leq \|\boldsymbol{\phi}\|_{\boldsymbol{\pi}} d\exp\left(kt^2(\gamma^2+b^2)\left(1+\frac{8}{1-\lambda}\right)\right).$$

Proving Lemma 1 is the technical core of our paper. The main idea is to write the expected trace expression in LHS of Lemma 1 in terms of the transition probability matrix $\boldsymbol{P}$, which allows for a recursive analysis to track how much the expected trace expression changes as a function of $k$. The analysis relies on incorporating the concentration of matrix-valued functions from [10] into the study of general Markov chains from [6], which was originally for scalars. Key to this extension is the definition of an inner product related to the stationary distribution $\boldsymbol{\pi}$ of $\boldsymbol{P}$, and a spectral expansion from such inner products. In contrast, the undirected regular graph case studied in [10] can be handled using the standard inner products, as well as the second largest eigenvalues of $\boldsymbol{P}$ instead of the spectral expansion. Detailed proofs of Theorem 3 and Lemma 1 can be found in Appendix B.2 and Appendix B.3 of the supplementary material, respectively.

Our result about real-valued matrices can be further generalized to complex-valued matrices, as stated in Theorem 1. Our main strategy is to adopt complexification technique [8], which first relate the eigenvalues of a $d \times d$ complex Hermitian matrix to a $2d \times 2d$ real symmetric matrix, and then deal with the real symmetric matrix using Theorem 3. The proof of Theorem 1 is deferred to Appendix B.4 in the supplementary material.

## 4 Convergence Rate of Co-occurrence Matrices of Markov Chains

In this section, we first apply the matrix Chernoff bound for regular Markov chains from Theorem 3 to obtain our main result on the convergence of co-occurrence matrix estimation, as stated in Theorem 2, and then discuss its generalization to Hidden Markov models in Corollary 1. Informally, our result in Theorem 3 states that if the mixing time of the Markov chain $\boldsymbol{P}$ is $\tau$, then the length of a trajectory needed to guarantee an additive error (in 2-norm) of $\epsilon$ is roughly $O\left((\tau+T)(\log n + \log \tau + T)/\epsilon^2 + T\right)$, where $T$ is the co-occurrence window size. However, we

cannot directly apply the matrix Chernoff bound because the co-occurrence matrix is not a sum of matrix-valued functions sampled from the original Markov chain $\boldsymbol{P}$. The main difficulty is to construct the proper Markov chain and matrix-valued function as desired by Theorem 3. We formally give our proof as follows:

*Proof.* (of Theorem 2) Our proof has three main steps: the first two construct a Markov chain $\boldsymbol{Q}$ according to $\boldsymbol{P}$, and a matrix-valued function $f$ such that the sums of matrix-valued random variables sampled via $\boldsymbol{Q}$ is exactly the error matrix $\boldsymbol{C} - \mathbb{AE}[\boldsymbol{C}]$. Then we invoke Theorem 3 to the constructed Markov chain $\boldsymbol{Q}$ and function $f$ to bound the convergence rate. We give details below.

**Step One** Given a random walk $(v_1, \cdots, v_L)$ on Markov chain $\boldsymbol{P}$, we construct a sequence $(X_1, \cdots, X_{L-T})$ where $X_i \triangleq (v_i, v_{i+1}, \cdots, v_{i+T})$, i.e., each $X_i$ is a size-$T$ sliding window over $(v_1, \cdots, v_L)$. Meanwhile, let $\mathcal{S}$ be the set of all $T$-step walks on Markov chain $\boldsymbol{P}$, we define a new Markov chain $\boldsymbol{Q}$ on $\mathcal{S}$ such that $\forall (u_0, \cdots, u_T), (w_0, \cdots, w_T) \in \mathcal{S}$:

$$\boldsymbol{Q}_{(u_0, \cdots, u_T), (w_0, \cdots, w_T)} \triangleq \begin{cases} \boldsymbol{P}_{u_T, w_T} & \text{if } (u_1, \cdots, u_T) = (w_0, \cdots, w_{T-1}); \\ 0 & \text{otherwise.} \end{cases}$$

The following claim characterizes the properties of $\boldsymbol{Q}$, whose proof is deferred to Appendix A.1 in the supplementary material.

**Claim 1** (Properties of $\boldsymbol{Q}$). *If $\boldsymbol{P}$ is a regular Markov chain, then $\boldsymbol{Q}$ satisfies:*

1. $\boldsymbol{Q}$ *is a regular Markov chain with stationary distribution* $\sigma_{(u_0, \cdots, u_T)} = \pi_{u_0} \boldsymbol{P}_{u_0, u_1} \cdots \boldsymbol{P}_{u_{T-1}, u_T}$;

2. *The sequence* $(X_1, \cdots X_{L-T})$ *is a random walk on $\boldsymbol{Q}$ starting from a distribution $\boldsymbol{\rho}$ such that* $\rho_{(u_0, \cdots, u_T)} = \phi_{u_0} \boldsymbol{P}_{u_0, u_1} \cdots \boldsymbol{P}_{u_{T-1}, u_T}$, *and* $\|\boldsymbol{\rho}\|_{\boldsymbol{\sigma}} = \|\boldsymbol{\phi}\|_{\boldsymbol{\pi}}$.

3. $\forall \delta > 0$, *the $\delta$-mixing time of $\boldsymbol{P}$ and $\boldsymbol{Q}$ satisfies* $\tau(\boldsymbol{Q}) < \tau(\boldsymbol{P}) + T$;

4. $\exists \boldsymbol{P}$ *with* $\lambda(\boldsymbol{P}) < 1$ *s.t. the induced $\boldsymbol{Q}$ has* $\lambda(\boldsymbol{Q}) = 1$, *i.e. $\boldsymbol{Q}$ may have zero spectral gap.*

Parts 1 and 2 imply that the sliding windows (i.e., $X_1, X_2, \cdots$) correspond to the state transition in a regular Markov chain $\boldsymbol{Q}$, whose mixing time and spectral expansion are described in Parts 3 and 4. A special case of the above construction when $T = 1$ can be found in Lemma 6.1 of [54].

**Step Two** Defining a matrix-valued function $f : \mathcal{S} \to \mathbb{R}^{n \times n}$ such that $\forall X = (u_0, \cdots, u_T) \in \mathcal{S}$:

$$f(X) \triangleq \frac{1}{2} \left( \sum_{r=1}^{T} \frac{\alpha_r}{2} \left( \boldsymbol{e}_{u_0} \boldsymbol{e}_{u_r}^\top + \boldsymbol{e}_{u_r} \boldsymbol{e}_{u_0}^\top \right) - \sum_{r=1}^{T} \frac{\alpha_r}{2} \left( \boldsymbol{\Pi} \boldsymbol{P}^r + (\boldsymbol{\Pi} \boldsymbol{P}^r)^\top \right) \right). \tag{2}$$

With this definition of $f(X)$, the difference between the co-occurrence matrix $\boldsymbol{C}$ and its asymptotic expectation $\mathbb{AE}[\boldsymbol{C}]$ can be written as: $\boldsymbol{C} - \mathbb{AE}[\boldsymbol{C}] = 2(\frac{1}{L-T} \sum_{i=1}^{L-T} f(X_i))$. We can further show the following properties of this function $f$:

**Claim 2** (Properties of $f$). *The function $f$ in Equation 2 satisfies (1) $\sum_{X \in \mathcal{S}} \sigma_X f(X) = 0$; (2) $f(X)$ is symmetric and $\|f(X)\|_2 \leq 1, \forall X \in \mathcal{S}$.*

This claim verifies that $f$ in Equation 2 satisfies the two conditions of matrix-valued function in Theorem 3. The proof of Claim 2 is deferred to Appendix A.2 of the supplementary material.

**Step Three** The construction in step two reveals the fact that the error matrix $\boldsymbol{C} - \mathbb{AE}[\boldsymbol{C}]$ can be written as the average of matrix-valued random variables (i.e., $f(X_i)$'s), which are sampled via a regular Markov chain $\boldsymbol{Q}$ This encourages us to directly apply Theorem 3. However, note that (1) the error probability in Theorem 3 contains a factor of spectral gap $(1 - \lambda)$; and (2) Part 4 of Claim 1 allows for the existence of a Markov chain $\boldsymbol{P}$ with $\lambda(\boldsymbol{P}) < 1$ while the induced Markov chain $\boldsymbol{Q}$ has $\lambda(\boldsymbol{Q}) = 1$. So we cannot directly apply Theorem 3 to $\boldsymbol{Q}$. To address this issue, we utilize the following tighter bound on sub-chains.

**Claim 3.** *(Claim 3.1 in Chung et al. [6]) Let $\boldsymbol{Q}$ be a regular Markov chain with $\delta$-mixing time $\tau(Q)$, then $\lambda\left(\boldsymbol{Q}^{\tau(Q)}\right) \leq \sqrt{2\delta}$. In particular, setting $\delta = \frac{1}{8}$ implies $\lambda(\boldsymbol{Q}^{\tau(\boldsymbol{Q})}) \leq \frac{1}{2}$.*

The above claim reveals the fact that, even though $\boldsymbol{Q}$ could have zero spectral gap (Part 4 of Claim 1), we can bound the spectral expansion of $\boldsymbol{Q}^{\tau(\boldsymbol{Q})}$. We partition $(X_1, \cdots X_{L-T})$ into $\tau(\boldsymbol{Q})$ groups[4], such that the $i$-th group consists of a sub-chain $(X_i, X_{i+\tau(\boldsymbol{Q})}, X_{i+2\tau(Q)}, \cdots)$ of length $k \triangleq (L-T)/\tau(\boldsymbol{Q})$. The sub-chain can be viewed as generated from a Markov chain $\boldsymbol{Q}^{\tau(Q)}$. Apply Theorem 3 to the $i$-th sub-chain, whose starting distribution is $\boldsymbol{\rho}_i \triangleq \left(\boldsymbol{Q}^\top\right)^{i-1} \boldsymbol{\rho}$, we have

$$\mathbb{P}\left[\lambda_{\max}\left(\frac{1}{k}\sum_{j=1}^{k} f(X_{i+(j-1)\tau(\boldsymbol{Q})})\right) \geq \epsilon\right] \leq \|\boldsymbol{\rho}_i\|_{\boldsymbol{\sigma}}\, n^2 \exp\left(-\epsilon^2\left(1-\lambda\left(\boldsymbol{Q}^{\tau(\boldsymbol{Q})}\right)\right)k/72\right)$$

$$\leq \|\boldsymbol{\rho}_i\|_{\boldsymbol{\sigma}}\, n^2 \exp\left(-\epsilon^2 k/144\right) \leq \|\boldsymbol{\phi}\|_{\boldsymbol{\pi}}\, n^2 \exp\left(-\epsilon^2 k/144\right),$$

where that last step follows by $\|\boldsymbol{\rho}_i\|_{\boldsymbol{\sigma}} \leq \|\boldsymbol{\rho}_{i-1}\|_{\boldsymbol{\sigma}} \leq \cdots \|\boldsymbol{\rho}_1\|_{\boldsymbol{\sigma}} = \|\boldsymbol{\rho}\|_{\boldsymbol{\sigma}}$ and $\|\boldsymbol{\rho}\|_{\boldsymbol{\sigma}} = \|\boldsymbol{\phi}\|_{\boldsymbol{\pi}}$ (Part 2 of Claim 1). Together with a union bound across each sub-chain, we can obtain:

$$\mathbb{P}\left[\lambda_{\max}\left(\boldsymbol{C} - \mathbb{A}\mathbb{E}[\boldsymbol{C}]\right) \geq \epsilon\right] = \mathbb{P}\left[\lambda_{\max}\left(\frac{1}{L-T}\sum_{j=1}^{L-T} f(X_j)\right) \geq \frac{\epsilon}{2}\right]$$

$$= \mathbb{P}\left[\lambda_{\max}\left(\frac{1}{\tau(\boldsymbol{Q})}\sum_{i=1}^{\tau(\boldsymbol{Q})}\frac{1}{k}\sum_{j=1}^{k} f(X_{i+(j-1)\tau(\boldsymbol{Q})})\right) \geq \frac{\epsilon}{2}\right]$$

$$\leq \sum_{i=1}^{\tau(\boldsymbol{Q})} \mathbb{P}\left[\lambda_{\max}\left(\frac{1}{k}\sum_{j=1}^{k} f(X_{i+(j-1)N})\right) \geq \frac{\epsilon}{2}\right] \leq \tau(\boldsymbol{Q})\, \|\boldsymbol{\phi}\|_{\boldsymbol{\pi}}\, n^2 \exp\left(-\epsilon^2 k/576\right).$$

The bound on $\lambda_{\min}$ also follows similarly. As $\boldsymbol{C} - \mathbb{A}\mathbb{E}[\boldsymbol{C}]$ is a real symmetric matrix, its 2-norm is its maximum absolute eigenvalue. Therefore, we can use the eigenvalue bound to bound the overall error probability in terms of the matrix 2-norm:

$$\mathbb{P}\left[\|\boldsymbol{C} - \mathbb{A}\mathbb{E}[\boldsymbol{C}]\|_2 \geq \epsilon\right] = \mathbb{P}\left[\lambda_{\max}(\boldsymbol{C} - \mathbb{A}\mathbb{E}[\boldsymbol{C}]) \geq \epsilon \vee \lambda_{\min}(\boldsymbol{C} - \mathbb{A}\mathbb{E}[\boldsymbol{C}]) \leq -\epsilon\right]$$

$$\leq 2\tau(\boldsymbol{Q})n^2\, \|\boldsymbol{\phi}\|_{\boldsymbol{\pi}}\, \exp\left(-\epsilon^2 k/576\right) \leq 2\left(\tau(\boldsymbol{P})+T\right)\|\boldsymbol{\phi}\|_{\boldsymbol{\pi}}\, n^2 \exp\left(-\frac{\epsilon^2(L-T)}{576\left(\tau(\boldsymbol{P})+T\right)}\right),$$

where the first inequality follows by union bound, and the second inequality is due to $\tau(\boldsymbol{Q}) < \tau(\boldsymbol{P}) + T$ (Part 3 of Claim 1). This bound implies that the probability that $\boldsymbol{C}$ deviates from $\mathbb{A}\mathbb{E}[\boldsymbol{C}]$ could be arbitrarily small by increasing the sampled trajectory length $L$. Specially, if we want the event $\|\boldsymbol{C} - \mathbb{A}\mathbb{E}[\boldsymbol{C}]\|_2 \geq \epsilon$ happens with probability smaller than $1/n^{O(1)}$, we need $L = O\left(\left(\tau(\boldsymbol{P})+T\right)\left(\log n + \log\left(\tau(\boldsymbol{P})+T\right)\right)/\epsilon^2 + T\right)$. If we assume $T = O(1)$, we can achieve $L = O\left(\tau(\boldsymbol{P})\left(\log n + \log\tau(\boldsymbol{P})\right)/\epsilon^2\right)$. $\qquad\square$

Our analysis can be extended to Hidden Markov models (HMM) as shown in Corollary 1, and has a potential to solve problems raised in [17, 30]. Our strategy is to treat the HMM with observable state space $\mathcal{Y}$ and hidden state space $\mathcal{X}$ as a Markov chain with state space $\mathcal{Y} \times \mathcal{X}$. The detailed proof can be found in Appendix A.3 in the supplementary material.

**Corollary 1** (Co-occurrence Matrices of HMMs). *For a HMM with observable states $y_t \in \mathcal{Y}$ and hidden states $x_t \in \mathcal{X}$, let $P(y_t|x_t)$ be the emission probability and $P(x_{t+1}|x_t)$ be the hidden state transition probability. Given an $L$-step trajectory observations from the HMM, $(y_1, \cdots, y_L)$, one needs a trajectory of length $L = O(\tau(\log|\mathcal{Y}| + \log\tau)/\epsilon^2)$ to achieve a co-occurrence matrix within error bound $\epsilon$ with high probability, where $\tau$ is the mixing time of the Markov chain on hidden states.*

## 5 Experiments

In this section, we show experiments to illustrate the exponentially fast convergence rate of estimating co-occurrence matrices of Markov chains. We conduct experiments on three synthetic Markov chains (Barbell graph, winning streak chain, and random graph) and one real-world Markov chain (BlogCatalog). For each Markov chain and each trajectory length $L$ from the set $\{10^1, \cdots, 10^7\}$, we measure the approximation error of the co-occurrence matrix constructed by Algorithm 1 from a $L$-step random walk sampled from the chain. We performed 64 trials for each

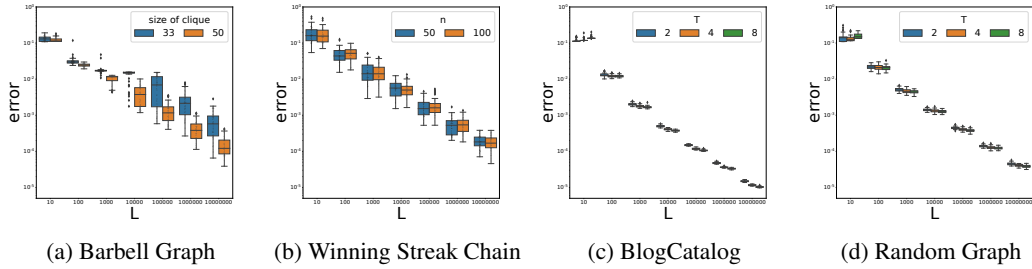

| (a) Barbell Graph | (b) Winning Streak Chain | (c) BlogCatalog | (d) Random Graph |

Figure 1: The convergence rate of co-occurrence matrices on Barbell graph, winning streak chain, BlogCatalog graph , and random graph (in log-log scale). The $x$-axis is the trajectory length $L$ and the $y$-axis is the error $\|C - \mathbb{AE}[C]\|_2$. Each experiment contains 64 trials, and the error bar is presented.

experiment, and the results are aggregated as an error-bar plot. We set $T = 2$ and $\alpha_r$ to be uniform unless otherwise mentioned. The relationship between trajectory length $L$ and approximation error $\|C - \mathbb{AE}[C]\|_2$ is shown in Figure 1 (in log-log scale). Across all the four datasets, the observed exponentially fast convergence rates match what our bounds predict in Theorem 2. Below we discuss our observations for each of these datasets.

**Barbell Graphs [43]** The Barbell graph is an undirected graph with two cliques connected by a single path. Such graphs' mixing times vary greatly: two cliques with size $k$ connected by a single edge have mixing time $\Theta(k^2)$; and two size-$k$ cliques connected by a length-$k$ path have mixing time about $\Theta(k^3)$. We evaluate the convergence rate of co-occurrence matrices on the two graphs mentioned above, each with 100 vertices. According to our bound that require $L = O(\tau(\log n + \log \tau)/\epsilon^2)$, we shall expect the approximate co-occurrence matrix to converge faster when the path bridging the two cliques is shorter. The experimental results are shown in Figure 1a, and indeed display faster convergences when the path is shorter (since we fix $n = 100$, a Barbell graph with clique size 50 has a shorter path connecting the two cliques than the one with clique size 33).

**Winning Streak Chains (Section 4.6 of [25])** A winning streak Markov chain has state space $[n]$, and can be viewed as tracking the number of consecutive 'tails' in a sequence of coin flips. Each state transits back to state 1 with probability 0.5, and the next state with probability 0.5. The $\delta$-mixing time of this chain satisfies $\tau \leq \lceil \log_2(1/\delta) \rceil$, and is independent of $n$. This prompted us to choose this chain, as we should expect similar rates of convergence for different values of $n$ according to our bound of $L = O(\tau(\log n + \log \tau)/\epsilon^2)$. In our experiment, we compare between $n = 50$ and $n = 100$ and illustrate the results in Figure 1b. As we can see, for each trajectory length $L$, the approximation errors of $n = 50$ and $n = 100$ are indeed very close.

**BlogCatalog Graph [47]** is widely used to benchmark graph representation learning algorithms [38, 12, 39]. It is an undirected graph of social relationships of online bloggers with 10,312 vertices and 333,983 edges. The random walk on the BlogCatalog graph has spectral expansion $\lambda \approx 0.57$. Following Levin and Peres [25], we can upper bound its $\frac{1}{8}$-mixing time by $\tau \leq 36$. We choose $T$ from $\{2, 4, 8\}$ and illustrate the results in Figure 1c. The convergence rate is robust to different values of $T$. Moreover, the variance in BlogCatalog is much smaller than that in other datasets.

We further demonstrate how our result could be used to select parameters for a popular graph representation learning algorithm, DeepWalk [38]. We set the window size $T = 10$, which is the default value of DeepWalk. Our bound on trajectory length $L$ in Theorem 1 (with explicit constant) is $L \geq 576(\tau + T)(3 \log n + \log(\tau + T))/\epsilon^2 + T$. The error bound $\epsilon$ might be chosen in the range of $[0.1, 0.01]$, which corresponds to $L$ in the range of $[8.4 \times 10^7, 8.4 \times 10^9]$. To verify that is a meaningful range for tuning $L$, we enumerate trajectory length $L$ from $\{10^4, \cdots, 10^{10}\}$, estimate the co-occurrence matrix with the single trajectory sampled from BlogCatalog, convert the co-occurrence matrix to the one implicitly factorized by DeepWalk [38, 39], and factorize it with SVD. For comparison, we also provide the result at the limiting case ($L \to +\infty$) where we directly compute the asymptotic expectation of the co-occurrence matrix according to Equation 1.

The limiting case involves computing a matrix polynomial and could be very expensive. For node classification task, the micro-F1 when training ratio is 50% is

| Length $L$ of DeepWalk | $10^4$ | $10^5$ | $10^6$ | $10^7$ | $10^8$ | $10^9$ | $10^{10}$ | $+\infty$ |
|---|---|---|---|---|---|---|---|---|
| Micro-F1 (%) | 15.21 | 18.31 | 26.99 | 33.85 | 39.12 | 41.28 | 41.58 | **41.82** |

.

As we can see, it is reasonable to choose $L$ in the predicted range.

**Random Graph** The small variance observed on BlogCatalog leads us to hypothesize that it shares some traits with random graphs. To gather further evidence for this, we estimate the co-occurrence matrices of an Erdős–Rényi random graph for comparison. Specifically, we take a random graph on 100 vertices where each undirected edge is added independently with probability 0.1, aka. $G(100, 0.1)$. The results Figure 1d show very similar behaviors compared to the BlogCatalog graph: small variance and robust convergence rates.

## 6 Conclusion and Future Work

In this paper, we analyze the convergence rate of estimating the co-occurrence matrix of a regular Markov chain. The main technical contribution of our work is to prove a Chernoff-type bound for sums of matrix-valued random variables sampled via a regular Markov chain, and we show that the problem of estimating co-occurrence matrices is a non-trivial application of the Chernoff-type bound. Our results show that, given a regular Markov chain with $n$ states and mixing time $\tau$, we need a trajectory of length $O(\tau(\log n + \log \tau)/\epsilon^2)$ to achieve an estimator of the co-occurrence matrix with error bound $\epsilon$. Our work leads to some natural future questions:

- Is it a tight bound? Our analysis on convergence rate of co-occurrence matrices relies on union bound, which probably gives a loose bound. It would be interesting to shave off the leading factor $\tau$ in the bound, as the mixing time $\tau$ could be large for some Markov chains.

- What if the construction of the co-occurrence matrix is coupled with a learning algorithm? For example, in word2vec [33], the co-occurrence in each sliding window outputs a mini-batch to a logistic matrix factorization model. This problem can be formalized as the convergence of stochastic gradient descent with non-i.i.d but Markovian random samples.

- Can we find more applications of the Markov chain matrix Chernoff bound? We believe Theorem 3 could have further applications, e.g., in reinforcement learning [36].

## Broader Impact

Our work contributes to the research literature of Chernoff-type bounds and co-occurrence statistics. Chernoff-type bound have become one of the most important probabilistic results in computer science. Our result generalize Chernoff bound to Markov dependence and random matrices. Co-occurrence statistics have emerged as important tools in machine learning. Our work addresses the sample complexity of estimating co-occurrence matrix. We believe such better theoretical understanding can further the understanding of potential and limitations of graph representation learning and reinforcement learning.

## Acknowledgments and Disclosure of Funding

We thank Jian Li (IIIS, Tsinghua) and Shengyu Zhang (Tencent Quantum Lab) for motivating this work. Funding in direct support of this work: Jiezhong Qiu and Jie Tang were supported by the National Key R&D Program of China (2018YFB1402600), NSFC for Distinguished Young Scholar (61825602), and NSFC (61836013). Richard Peng was partially supported by NSF grant CCF-1846218. There is no additional revenue related to this work.

## Footnotes

[1]Please note that regular Markov chains are Markov chains which are aperiodic and irreducible, while an undirected regular graph is an undirected graph where each vertex has the same number of neighbors. In this work, the term "regular" may have different meanings depending on the context.

[2]The volume of a graph $G$ is defined to be $\text{vol}(G) \triangleq \sum_i \sum_j \boldsymbol{A}_{ij}$.

[3] Please note that we need the Markov chain to be regular to make the mixing-time well-defined. For an ergodic Markov chain which could be periodic, the mixing time may be ill-defined.

[4]Without loss of generality, we assume $L - T$ is a multiple of $\tau(\boldsymbol{Q})$.

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
