[Supplementary Material]

# Supplementary Material of A Matrix Chernoff Bound for Markov Chains and Its Application to Co-occurrence Matrices

## A Convergence Rate of Co-occurrence Matrices

### A.1 Proof of Claim 1

**Claim 1** (Properties of $\boldsymbol{Q}$). *If $\boldsymbol{P}$ is a regular Markov chain, then $\boldsymbol{Q}$ satisfies:*

1. $\boldsymbol{Q}$ *is a regular Markov chain with stationary distribution* $\sigma_{(u_0,\cdots,u_T)} = \pi_{u_0}\boldsymbol{P}_{u_0,u_1}\cdots\boldsymbol{P}_{u_{T-1},u_T}$;

2. *The sequence* $(X_1,\cdots X_{L-T})$ *is a random walk on* $\boldsymbol{Q}$ *starting from a distribution* $\boldsymbol{\rho}$ *such that* $\rho_{(u_0,\cdots,u_T)} = \phi_{u_0}\boldsymbol{P}_{u_0,u_1}\cdots\boldsymbol{P}_{u_{T-1},u_T}$, *and* $\|\boldsymbol{\rho}\|_{\boldsymbol{\sigma}} = \|\boldsymbol{\phi}\|_{\boldsymbol{\pi}}$.

3. $\forall\delta > 0$, *the $\delta$-mixing time of $\boldsymbol{P}$ and $\boldsymbol{Q}$ satisfies* $\tau(\boldsymbol{Q}) < \tau(\boldsymbol{P}) + T$;

4. $\exists\boldsymbol{P}$ *with* $\lambda(\boldsymbol{P}) < 1$ *s.t. the induced $\boldsymbol{Q}$ has* $\lambda(\boldsymbol{Q}) = 1$, *i.e. $\boldsymbol{Q}$ may have zero spectral gap.*

*Proof.* We prove the fours parts of this Claim one by one.

**Part 1** To prove $\boldsymbol{Q}$ is regular, it is sufficient to show that $\exists N_1, \forall n_1 > N_1, (v_0,\cdots,v_T)$ can reach $(u_0,\cdots,u_T)$ at $n_1$ steps. We know $\boldsymbol{P}$ is a regular Markov chain, so there exists $N_2 \geq T$ s.t., for any $n_2 \geq N_2$, $v_T$ can reach $u_0$ at exact $n_2$ step, i,e., there is a $n_2$-step walk s.t. $(v_T, w_1, \cdots, w_{n_2-1}, u_0)$ on $\boldsymbol{P}$. This induces an $n_2$-step walk from $(v_0,\cdots,v_T)$ to $(w_{n_2-T+1},\cdots,w_{n_2-1},u_0)$. Take further $T$ step, we can reach $(u_0,\cdots,u_T)$, so we construct a $n_1 = n_2 + T$ step walk from $(v_0,\cdots,v_T)$ to $(u_0,\cdots u_T)$. Since this is true for any $n_2 \geq N_2$, we then claim that any state can be reached from any other state in any number of steps greater than or equal to a number $N_1 = N_2 + T$. Next to verify $\boldsymbol{\sigma}$ such that $\sigma_{(u_0,\cdots,u_T)} = \pi_{u_0}\boldsymbol{P}_{u_0,u_1}\cdots\boldsymbol{P}_{u_{T-1},u_T}$ is the stationary distribution of Markov chain $\boldsymbol{Q}$,

$$\sum_{(u_0,\cdots,u_T)\in\mathcal{S}} \sigma_{(u_0,\cdots,u_T)}\boldsymbol{Q}_{(u_0,\cdots,u_T),(w_0,\cdots,w_T)}$$

$$= \sum_{u_0:(u_0,w_0,\cdots,w_{T-1})\in\mathcal{S}} \pi_{u_0}\boldsymbol{P}_{u_0,w_0}\boldsymbol{P}_{w_0,w_1},\cdots,\boldsymbol{P}_{w_{T-2},w_{T-1}}\boldsymbol{P}_{w_{T-1},w_T}$$

$$= \left(\sum_{u_0}\pi_{u_0}\boldsymbol{P}_{u_0,w_0}\right)\boldsymbol{P}_{w_0,w_1},\cdots,\boldsymbol{P}_{w_{T-2},w_{T-1}}\boldsymbol{P}_{w_{T-1},w_T}$$

$$= \pi_{w_0}\boldsymbol{P}_{w_0,w_1},\cdots,\boldsymbol{P}_{w_{T-2},w_{T-1}}\boldsymbol{P}_{w_{T-1},w_T} = \sigma_{w_0,\cdots,w_T}.$$

**Part 2** Recall $(v_1,\cdots,v_L)$ is a random walk on $\boldsymbol{P}$ starting from distribution $\phi$, so the probability we observe $X_1 = (v_1,\cdots,v_{T+1})$ is $\phi_{v_1}\boldsymbol{P}_{v_1,v_2}\cdots\boldsymbol{P}_{v_T,v_T} = \rho_{(v_1,\cdots,v_{T+1})}$, i.e., $X_1$ is sampled from the distribution $\boldsymbol{\rho}$. Then we study the transition probability from $X_i = (v_i,\cdots,v_{i+T})$ to $X_{i+1} = (v_{i+1},\cdots,v_{i+T+1})$, which is $\boldsymbol{P}_{v_{i+T},v_{i+T+1}} = \boldsymbol{Q}_{X_i,X_{i+1}}$. Consequently, we can claim $(X_i,\cdots,X_{L-T})$ is a random walk on $\boldsymbol{Q}$. Moreover,

$$\|\boldsymbol{\rho}\|_{\boldsymbol{\sigma}}^2 = \sum_{(u_0,\cdots,u_T)\in\mathcal{S}} \frac{\rho_{(u_0,\cdots,u_T)}^2}{\sigma_{(u_0,\cdots,u_T)}} = \sum_{(u_0,\cdots,u_T)\in\mathcal{S}} \frac{\left(\phi_{u_0}\boldsymbol{P}_{u_0,u_1}\cdots\boldsymbol{P}_{u_{T-1},u_T}\right)^2}{\pi_{u_0}\boldsymbol{P}_{u_0,u_1}\cdots\boldsymbol{P}_{u_{T-1},u_T}}$$

$$= \sum_{u_0}\frac{\phi_{u_0}^2}{\pi_{u_0}}\sum_{(u_0,u_1,\cdots,u_T)\in\mathcal{S}}\boldsymbol{P}_{u_0,u_1}\cdots\boldsymbol{P}_{u_{T-1},u_T} = \sum_{u_0}\frac{\phi_{u_0}^2}{\pi_{u_0}} = \|\boldsymbol{\phi}\|_{\boldsymbol{\pi}}^2,$$

which implies $\|\boldsymbol{\rho}\|_{\boldsymbol{\sigma}} = \|\boldsymbol{\phi}\|_{\boldsymbol{\pi}}$.

**Part 3** For any distribution $\boldsymbol{y}$ on $\mathcal{S}$, define $\boldsymbol{x} \in \mathbb{R}^n$ such that $x_i = \sum_{(v_1,\cdots,v_{T-1},i)\in\mathcal{S}} y_{v_1,\cdots,v_{T-1},i}$. Easy to see $\boldsymbol{x}$ is a probability vector, since $\boldsymbol{x}$ is the marginal probability of $\boldsymbol{y}$. For convenience, we

assume for a moment the $\boldsymbol{x}, \boldsymbol{y}, \boldsymbol{\sigma}, \boldsymbol{\pi}$ are row vectors. We can see that:

$$
\begin{aligned}
\left\|\boldsymbol{y}\boldsymbol{Q}^{\tau(\boldsymbol{P})+T-1} - \boldsymbol{\sigma}\right\|_{TV} &= \frac{1}{2}\left\|\boldsymbol{y}\boldsymbol{Q}^{\tau(\boldsymbol{P})+T-1} - \boldsymbol{\sigma}\right\|_1 \\
&= \frac{1}{2}\sum_{(v_1,\cdots,v_T)\in\mathcal{S}}\left|\left(\boldsymbol{y}\boldsymbol{Q}^{\tau(\boldsymbol{P})+T-1} - \boldsymbol{\sigma}\right)_{v_1,\cdots,v_T}\right| \\
&= \frac{1}{2}\sum_{(v_1,\cdots,v_T)\in\mathcal{S}}\left|\left(\boldsymbol{x}\boldsymbol{P}^{\tau(\boldsymbol{P})}\right)_{v_1}\boldsymbol{P}_{v_1,v_2}\cdots\boldsymbol{P}_{v_{T-1},v_T} - \boldsymbol{\pi}_{v_1}\boldsymbol{P}_{v_1,v_2}\cdots\boldsymbol{P}_{v_{T-1},v_T}\right| \\
&= \frac{1}{2}\sum_{(v_1,\cdots,v_T)\in\mathcal{S}}\left|\left(\boldsymbol{x}\boldsymbol{P}^{\tau(\boldsymbol{P})}\right)_{v_1} - \pi_{v_1}\right|\boldsymbol{P}_{v_1,v_2}\cdots\boldsymbol{P}_{v_{T-1},v_T} \\
&= \frac{1}{2}\sum_{v_1}\left|\left(\boldsymbol{x}\boldsymbol{P}^{\tau(\boldsymbol{P})}\right)_{v_1} - \pi_{v_1}\right|\sum_{(v_1,\cdots,v_T)\in\mathcal{S}}\boldsymbol{P}_{v_1,v_2}\cdots\boldsymbol{P}_{v_{T-1},v_T} \\
&= \frac{1}{2}\sum_{v_1}\left|\left(\boldsymbol{x}\boldsymbol{P}^{\tau(\boldsymbol{P})}\right)_{v_1} - \pi_{v_1}\right| = \frac{1}{2}\left\|\boldsymbol{x}\boldsymbol{P}^{\tau(\boldsymbol{P})} - \boldsymbol{\pi}\right\|_1 = \left\|\boldsymbol{x}\boldsymbol{P}^{\tau(\boldsymbol{P})} - \boldsymbol{\pi}\right\|_{TV} \le \delta.
\end{aligned}
$$

which indicates $\tau(\boldsymbol{Q}) \le \tau(\boldsymbol{P}) + T - 1 < \tau(\boldsymbol{P}) + T$.

**Part 4** This is an example showing that $\lambda(\boldsymbol{Q})$ cannot be bounded by $\lambda(\boldsymbol{P})$ — even though $\boldsymbol{P}$ has $\lambda(\boldsymbol{P}) < 1$, the induced $\boldsymbol{Q}$ may have $\lambda(\boldsymbol{Q}) = 1$. We consider random walk on the unweighted undirected graph [image] and $T = 1$. The transition probability matrix $\boldsymbol{P}$ is:

$$
\boldsymbol{P} = \begin{bmatrix} 0 & 1/3 & 1/3 & 1/3 \\ 1/2 & 0 & 1/2 & 0 \\ 1/3 & 1/3 & 0 & 1/3 \\ 1/2 & 0 & 1/2 & 0 \end{bmatrix}
$$

with stationary distribution $\boldsymbol{\pi} = \begin{bmatrix} 0.3 & 0.2 & 0.3 & 0.2 \end{bmatrix}^\top$ and $\lambda(\boldsymbol{P}) = \frac{2}{3}$. When $T = 1$, the induced Markov chain $\boldsymbol{Q}$ has stationary distribution $\sigma_{u,v} = \pi_u \boldsymbol{P}_{u,v} = \frac{d_u}{2m}\frac{1}{d_u} = \frac{1}{2m}$ where $m = 5$ is the number of edges in the graph. Construct $\boldsymbol{y} \in \mathbb{R}^{|\mathcal{S}|}$ such that

$$
y_{(u,v)} = \begin{cases} 1 & (u,v) = (0,1), \\ -1 & (u,v) = (0,3), \\ 0 & \text{otherwise.} \end{cases}
$$

The constructed vector $\boldsymbol{y}$ has norm

$$
\|\boldsymbol{y}\|_\sigma = \sqrt{\langle \boldsymbol{y}, \boldsymbol{y}\rangle_\sigma} = \sqrt{\sum_{(u,v)\in\mathcal{S}}\frac{y_{(u,v)}y_{(u,v)}}{\sigma_{(u,v)}}} = \sqrt{\frac{y_{(0,1)}y_{(0,1)}}{\sigma_{(0,1)}} + \frac{y_{(0,3)}y_{(0,3)}}{\sigma_{(0,3)}}} = 2\sqrt{m}.
$$

And it is easy to check $\boldsymbol{y} \perp \boldsymbol{\sigma}$, since $\langle \boldsymbol{y}, \boldsymbol{\sigma}\rangle_\sigma = \sum_{(u,v)\in\mathcal{S}}\frac{\sigma_{(u,v)}y_{(u,v)}}{\sigma_{(u,v)}} = y_{(0,1)} + y_{(0,3)} = 0$. Let $\boldsymbol{x} = (\boldsymbol{y}^*\boldsymbol{Q})^*$, we have for $(u,v) \in \mathcal{S}$:

$$
\boldsymbol{x}_{(u,v)} = \begin{cases} 1 & (u,v) = (1,2), \\ -1 & (u,v) = (3,2), \\ 0 & \text{otherwise.} \end{cases}
$$

This vector has norm:

$$
\|\boldsymbol{x}\|_\sigma = \sqrt{\langle \boldsymbol{x}, \boldsymbol{x}\rangle_\sigma} = \sqrt{\sum_{(u,v)\in\mathcal{S}}\frac{x_{(u,v)}x_{(u,v)}}{\sigma_{(u,v)}}} = \sqrt{\frac{y_{(1,2)}y_{(1,2)}}{\sigma_{(1,2)}} + \frac{y_{(3,2)}y_{(3,2)}}{\sigma_{(3,2)}}} = 2\sqrt{m}
$$

Thus we have $\frac{\|(\boldsymbol{y}^*\boldsymbol{Q})^*\|_\sigma}{\|\boldsymbol{y}\|_\sigma} = 1$. Taking maximum over all possible $\boldsymbol{y}$ gives $\lambda(\boldsymbol{Q}) \ge 1$. Also note that fact that $\lambda(\boldsymbol{Q}) \le 1$, so $\lambda(\boldsymbol{Q}) = 1$. $\qquad\square$

## A.2 Proof of Claim 2

**Claim 2** (Properties of $f$). *The function $f$ in Equation 2 satisfies (1) $\sum_{X\in\mathcal{S}}\sigma_X f(X) = 0$; (2) $f(X)$ is symmetric and $\|f(X)\|_2 \le 1, \forall X \in \mathcal{S}$.*

*Proof.* Note that Equation 2 is indeed a random value minus its expectation, so naturally Equation 2 has zero mean, i.e., $\sum_{X \in \mathcal{S}} \sigma_X f(X) = 0$. Moreover, $\|f(X)\|_2 \leq 1$ because

$$\|f(X)\|_2 \leq \frac{1}{2} \left( \sum_{r=1}^{T} \frac{|\alpha_r|}{2} \left( \left\| \boldsymbol{e}_{v_0} \boldsymbol{e}_{v_r}^\top \right\|_2 + \left\| \boldsymbol{e}_{v_r} \boldsymbol{e}_{v_0}^\top \right\|_2 \right) + \sum_{r=1}^{T} \frac{|\alpha_r|}{2} \left( \|\boldsymbol{\Pi}\|_2 \|\boldsymbol{P}\|_2^r + \left\| \boldsymbol{P}^\top \right\|_2^r \|\boldsymbol{\Pi}\|_2 \right) \right)$$

$$\leq \frac{1}{2} \left( \sum_{r=1}^{T} |\alpha_r| + \sum_{r=1}^{T} |\alpha_r| \right) = 1.$$

where the first step follows triangle inequality and submultiplicativity of 2-norm, and the third step follows by (1) $\left\| \boldsymbol{e}_i \boldsymbol{e}_j^\top \right\|_2 = 1$; (2) $\|\boldsymbol{\Pi}\|_2 = \|\operatorname{diag}(\boldsymbol{\pi})\|_2 \leq 1$ for distribution $\boldsymbol{\pi}$; (3) $\|\boldsymbol{P}\|_2 = \left\| \boldsymbol{P}^\top \right\|_2 = 1$. $\square$

### A.3 Proof of Corollary 1

**Corollary 1** (Co-occurrence Matrices of HMMs). *For a HMM with observable states $y_t \in \mathcal{Y}$ and hidden states $x_t \in \mathcal{X}$, let $P(y_t|x_t)$ be the emission probability and $P(x_{t+1}|x_t)$ be the hidden state transition probability. Given an $L$-step trajectory observations from the HMM, $(y_1, \cdots, y_L)$, one needs a trajectory of length $L = O(\tau(\log|\mathcal{Y}| + \log \tau)/\epsilon^2)$ to achieve a co-occurrence matrix within error bound $\epsilon$ with high probability, where $\tau$ is the mixing time of the Markov chain on hidden states.*

*Proof.* A HMM can be model by a Markov chain $\boldsymbol{P}$ on $\mathcal{Y} \times \mathcal{X}$ such that $P(y_{t+1}, x_{t+1}|y_t, x_t) = P(y_{t+1}|x_{t+1})P(x_{t+1}|x_t)$. For the co-occurrence matrix of observable states, applying a similar proof like our Theorem 2 shows that one needs a trajectory of length $O(\tau(\boldsymbol{P})(\log|\mathcal{Y}| + \log \tau(\boldsymbol{P}))/\epsilon^2)$ to achieve error bound $\epsilon$ with high probability. Moreover, the mixing time $\tau(\boldsymbol{P})$ is bounded by the mixing time of the Markov chain on the hidden state space (i.e., $P(x_{t+1}|x_t)$). $\square$

## B Matrix Chernoff Bounds for Markov Chains

### B.1 Preliminaries

**Kronecker Products** If $\boldsymbol{A}$ is an $M_1 \times N_1$ matrix and $\boldsymbol{B}$ is a $M_2 \times N_2$ matrix, then the Kronecker product $\boldsymbol{A} \otimes \boldsymbol{B}$ is the $M_2 M_1 \times N_1 N_2$ block matrix such that

$$\boldsymbol{A} \otimes \boldsymbol{B} = \begin{bmatrix} \boldsymbol{A}_{1,1} \boldsymbol{B} & \cdots & \boldsymbol{A}_{1,N_1} \boldsymbol{B} \\ \vdots & \ddots & \vdots \\ \boldsymbol{A}_{M_1,1} \boldsymbol{B} & \cdots & \boldsymbol{A}_{M_1,N_1} \boldsymbol{B} \end{bmatrix}.$$

Kronecker product has the mixed-product property. If $\boldsymbol{A}, \boldsymbol{B}, \boldsymbol{C}, \boldsymbol{D}$ are matrices of such size that one can from the matrix products $\boldsymbol{AC}$ and $\boldsymbol{BD}$, then $(\boldsymbol{A} \otimes \boldsymbol{B})(\boldsymbol{C} \otimes \boldsymbol{D}) = (\boldsymbol{AC}) \otimes (\boldsymbol{BD})$.

**Vectorization** For a matrix $\boldsymbol{X} \in \mathbb{C}^{d \times d}$, $\operatorname{vec}(\boldsymbol{X}) \in \mathbb{C}^{d^2}$ denote the vertorization of the matrix $\boldsymbol{X}$, s.t. $\operatorname{vec}(\boldsymbol{X}) = \sum_{i \in [d]} \sum_{j \in [d]} \boldsymbol{X}_{i,j} \boldsymbol{e}_i \otimes \boldsymbol{e}_j$, which is the stack of rows of $\boldsymbol{X}$. And there is a relationship between matrix multiplication and Kronecker product s.t. $\operatorname{vec}(\boldsymbol{AXB}) = (\boldsymbol{A} \otimes \boldsymbol{B}^\top) \operatorname{vec}(\boldsymbol{X})$.

**Matrices and Norms** For a matrix $\boldsymbol{A} \in \mathbb{C}^{N \times N}$, we use $\boldsymbol{A}^\top$ to denote matrix transpose, use $\overline{\boldsymbol{A}}$ to denote entry-wise matrix conjugation, use $\boldsymbol{A}^*$ to denote matrix conjugate transpose ($\boldsymbol{A}^* = \overline{\boldsymbol{A}^\top} = \overline{\boldsymbol{A}}^\top$). The vector 2-norm is defined to be $\|\boldsymbol{x}\|_2 = \sqrt{\boldsymbol{x}^* \boldsymbol{x}}$, and the matrix 2-norm is defined to be $\|\boldsymbol{A}\|_2 = \max_{\boldsymbol{x} \in \mathbb{C}^N, \boldsymbol{x} \neq 0} \frac{\|\boldsymbol{Ax}\|_2}{\|\boldsymbol{x}\|_2}$.

We then recall the definition of inner-product under $\boldsymbol{\pi}$-kernel in Section 2. The inner-product under $\boldsymbol{\pi}$-kernel for $\mathbb{C}^N$ is $\langle \boldsymbol{x}, \boldsymbol{y} \rangle_{\boldsymbol{\pi}} = \boldsymbol{y}^* \boldsymbol{\Pi}^{-1} \boldsymbol{x}$ where $\boldsymbol{\Pi} = \operatorname{diag}(\boldsymbol{\pi})$, and its induced $\boldsymbol{\pi}$-norm $\|\boldsymbol{x}\|_{\boldsymbol{\pi}} = \sqrt{\langle \boldsymbol{x}, \boldsymbol{x} \rangle_{\boldsymbol{\pi}}}$. The above definition allow us to define a inner product under $\boldsymbol{\pi}$-kernel on $\mathbb{C}^{Nd^2}$:

**Definition 1.** *Define inner product on $\mathbb{C}^{Nd^2}$ under $\boldsymbol{\pi}$-kernel to be $\langle \boldsymbol{x}, \boldsymbol{y} \rangle_{\boldsymbol{\pi}} = \boldsymbol{y}^* \left( \boldsymbol{\Pi}^{-1} \otimes \boldsymbol{I}_{d^2} \right) \boldsymbol{x}$.*

**Remark 1.** *For $\boldsymbol{x}, \boldsymbol{y} \in \mathbb{C}^N$ and $\boldsymbol{p}, \boldsymbol{q} \in \mathbb{C}^{d^2}$, then inner product (under $\boldsymbol{\pi}$-kernel) between $\boldsymbol{x} \otimes \boldsymbol{p}$ and $\boldsymbol{y} \otimes \boldsymbol{q}$ can be simplified as*

$$\langle \boldsymbol{x} \otimes \boldsymbol{p}, \boldsymbol{y} \otimes \boldsymbol{q} \rangle_{\boldsymbol{\pi}} = (\boldsymbol{y} \otimes \boldsymbol{q})^* \left( \boldsymbol{\Pi}^{-1} \otimes \boldsymbol{I}_{d^2} \right) (\boldsymbol{x} \otimes \boldsymbol{p}) = (\boldsymbol{y}^* \boldsymbol{\Pi}^{-1} \boldsymbol{x}) \otimes (\boldsymbol{q}^* \boldsymbol{p}) = \langle \boldsymbol{x}, \boldsymbol{y} \rangle_{\boldsymbol{\pi}} \langle \boldsymbol{p}, \boldsymbol{q} \rangle.$$

**Remark 2.** *The induced $\pi$-norm is $\|x\|_\pi = \sqrt{\langle x, x\rangle_\pi}$. When $x = y \otimes w$, the $\pi$-norm can be simplified to be: $\|x\|_\pi = \sqrt{\langle y \otimes w, y \otimes w\rangle_\pi} = \sqrt{\langle y, y\rangle_\pi \langle w, w\rangle} = \|y\|_\pi \|w\|_2$.*

**Matrix Exponential** The matrix exponential of a matrix $A \in \mathbb{C}^{d \times d}$ is defined by Taylor expansion $\exp(A) = \sum_{j=0}^{+\infty} \frac{A^j}{j!}$. And we will use the fact that $\exp(A) \otimes \exp(B) = \exp(A \otimes I + I \otimes B)$.

**Golden-Thompson Inequality** We need the following multi-matrix Golden-Thompson inequality from from Garg et al. [10].

**Theorem 4** (Multi-matrix Golden-Thompson Inequality, Theorem 1.5 in [10]). *Let $H_1, \cdots H_k$ be $k$ Hermitian matrices, then for some probability distribution $\mu$ on $[-\frac{\pi}{2}, \frac{\pi}{2}]$.*

$$\log\left(\mathrm{Tr}\left[\exp\left(\sum_{j=1}^k H_j\right)\right]\right) \leq \frac{4}{\pi}\int_{-\frac{\pi}{2}}^{\frac{\pi}{2}} \log\left(\mathrm{Tr}\left[\prod_{j=1}^k \exp\left(\frac{e^{\mathrm{i}\phi}}{2}H_j\right)\prod_{j=k}^1 \exp\left(\frac{e^{-\mathrm{i}\phi}}{2}H_j\right)\right]\right)d\mu(\phi).$$

## B.2 Proof of Theorem 3

**Theorem 3** (A Real-Valued Version of Theorem 1). *Let $P$ be a regular Markov chain with state space $[N]$, stationary distribution $\pi$ and spectral expansion $\lambda$. Let $f : [N] \to \mathbb{R}^{d \times d}$ be a function such that (1) $\forall v \in [N]$, $f(v)$ is symmetric and $\|f(v)\|_2 \leq 1$; (2) $\sum_{v \in [N]} \pi_v f(v) = 0$. Let $(v_1, \cdots, v_k)$ denote a $k$-step random walk on $P$ starting from a distribution $\phi$ on $[N]$. Then given $\epsilon \in (0, 1)$,*

$$\mathbb{P}\left[\lambda_{\max}\left(\frac{1}{k}\sum_{j=1}^k f(v_j)\right) \geq \epsilon\right] \leq \|\phi\|_\pi d^2 \exp\left(-(\epsilon^2(1-\lambda)k/72)\right)$$

$$\mathbb{P}\left[\lambda_{\min}\left(\frac{1}{k}\sum_{j=1}^k f(v_j)\right) \leq -\epsilon\right] \leq \|\phi\|_\pi d^2 \exp\left(-(\epsilon^2(1-\lambda)k/72)\right).$$

*Proof.* Due to symmetry, it suffices to prove one of the statements. Let $t > 0$ be a parameter to be chosen later. Then

$$\mathbb{P}\left[\lambda_{\max}\left(\frac{1}{k}\sum_{j=1}^k f(v_j)\right) \geq \epsilon\right] = \mathbb{P}\left[\lambda_{\max}\left(\sum_{j=1}^k f(v_j)\right) \geq k\epsilon\right]$$

$$\leq \mathbb{P}\left[\mathrm{Tr}\left[\exp\left(t\sum_{j=1}^k f(v_j)\right)\right] \geq \exp(tk\epsilon)\right] \qquad (3)$$

$$\leq \frac{\mathbb{E}_{v_1\cdots,v_k}\left[\mathrm{Tr}\left[\exp\left(t\sum_{j=1}^k f(v_j)\right)\right]\right]}{\exp(tk\epsilon)}.$$

The second inequality follows Markov inequality.

Next to bound $\mathbb{E}_{v_1\cdots,v_k}\left[\mathrm{Tr}\left[\exp\left(t\sum_{j=1}^k f(v_j)\right)\right]\right]$. Using Theorem 4, we have:

$$\log\left(\mathrm{Tr}\left[\exp\left(t\sum_{j=1}^k f(v_j)\right)\right]\right) \leq \frac{4}{\pi}\int_{-\frac{\pi}{2}}^{\frac{\pi}{2}}\log\left(\mathrm{Tr}\left[\prod_{j=1}^k \exp\left(\frac{e^{\mathrm{i}\phi}}{2}tf(v_j)\right)\prod_{j=k}^1 \exp\left(\frac{e^{-\mathrm{i}\phi}}{2}tf(v_j)\right)\right]\right)d\mu(\phi)$$

$$\leq \frac{4}{\pi}\log\int_{-\frac{\pi}{2}}^{\frac{\pi}{2}}\mathrm{Tr}\left[\prod_{j=1}^k \exp\left(\frac{e^{\mathrm{i}\phi}}{2}tf(v_j)\right)\prod_{j=k}^1 \exp\left(\frac{e^{-\mathrm{i}\phi}}{2}tf(v_j)\right)\right]d\mu(\phi),$$

where the second step follows by concavity of $\log$ function and the fact that $\mu(\phi)$ is a probability distribution on $[-\frac{\pi}{2}, \frac{\pi}{2}]$. This implies

$$\mathrm{Tr}\left[\exp\left(t\sum_{j=1}^k f(v_j)\right)\right] \leq \left(\int_{-\frac{\pi}{2}}^{\frac{\pi}{2}}\mathrm{Tr}\left[\prod_{j=1}^k \exp\left(\frac{e^{\mathrm{i}\phi}}{2}tf(v_j)\right)\prod_{j=k}^1 \exp\left(\frac{e^{-\mathrm{i}\phi}}{2}tf(v_j)\right)\right]d\mu(\phi)\right)^{\frac{4}{\pi}}.$$

Note that $\|x\|_p \leq d^{1/p-1}\|x\|_1$ for $p \in (0, 1)$, choosing $p = \pi/4$ we have

$$\left(\mathrm{Tr}\left[\exp\left(\frac{\pi}{4}t\sum_{j=1}^k f(v_j)\right)\right]\right)^{\frac{4}{\pi}} \leq d^{\frac{4}{\pi}-1}\mathrm{Tr}\left[\exp\left(t\sum_{j=1}^k f(v_j)\right)\right].$$

Combining the above two equations together, we have

$$\mathrm{Tr}\left[\exp\left(\frac{\pi}{4}t\sum_{j=1}^{k}f(v_j)\right)\right] \le d^{1-\frac{\pi}{4}}\int_{-\frac{\pi}{2}}^{\frac{\pi}{2}}\mathrm{Tr}\left[\prod_{j=1}^{k}\exp\left(\frac{e^{\mathrm{i}\phi}}{2}tf(v_j)\right)\prod_{j=k}^{1}\exp\left(\frac{e^{-\mathrm{i}\phi}}{2}tf(v_j)\right)\right]d\mu(\phi). \quad (4)$$

Write $e^{\mathrm{i}\phi} = \gamma + \mathrm{i}b$ with $\gamma^2 + b^2 = |\gamma + \mathrm{i}b|^2 = |e^{\mathrm{i}\phi}|^2 = 1$:

**Lemma 1** (Analogous to Lemma 4.3 in [10]). *Let $P$ be a regular Markov chain with state space $[N]$ with spectral expansion $\lambda$. Let $f$ be a function $f : [N] \to \mathbb{R}^{d\times d}$ such that (1) $\sum_{v\in[N]}\pi_v f(v) = 0$; (2) $\|f(v)\|_2 \le 1$ and $f(v)$ is symmetric, $v \in [N]$. Let $(v_1, \cdots, v_k)$ denote a k-step random walk on $P$ starting from a distribution $\phi$ on $[N]$. Then for any $t > 0, \gamma \ge 0, b > 0$ such that $t^2(\gamma^2 + b^2) \le 1$ and $t\sqrt{\gamma^2 + b^2} \le \frac{1-\lambda}{4\lambda}$, we have*

$$\mathbb{E}\left[\mathrm{Tr}\left[\prod_{j=1}^{k}\exp\left(\frac{tf(v_j)(\gamma+\mathrm{i}b)}{2}\right)\prod_{j=k}^{1}\exp\left(\frac{tf(v_j)(\gamma-\mathrm{i}b)}{2}\right)\right]\right] \le \|\phi\|_\pi\, d\exp\left(kt^2(\gamma^2+b^2)\left(1+\frac{8}{1-\lambda}\right)\right).$$

Assuming the above lemma, we can complete the proof of the theorem as:

$$\mathbb{E}_{v_1\cdots,v_k}\left[\mathrm{Tr}\left[\exp\left(\frac{\pi}{4}t\sum_{j=1}^{k}f(v_j)\right)\right]\right]$$

$$\le d^{1-\frac{\pi}{4}}\mathbb{E}_{v_1\cdots,v_k}\left[\int_{-\frac{\pi}{2}}^{\frac{\pi}{2}}\left(\mathrm{Tr}\left[\prod_{j=1}^{k}\exp\left(\frac{e^{\mathrm{i}\phi}}{2}tf(v_j)\right)\prod_{j=k}^{1}\exp\left(\frac{e^{-\mathrm{i}\phi}}{2}tf(v_j)\right)\right]\right)d\mu(\phi)\right]$$

$$= d^{1-\frac{\pi}{4}}\int_{-\frac{\pi}{2}}^{\frac{\pi}{2}}\mathbb{E}_{v_1\cdots,v_k}\left[\mathrm{Tr}\left[\prod_{j=1}^{k}\exp\left(\frac{e^{\mathrm{i}\phi}}{2}tf(v_j)\right)\prod_{j=k}^{1}\exp\left(\frac{e^{-\mathrm{i}\phi}}{2}tf(v_j)\right)\right]\right]d\mu(\phi) \quad (5)$$

$$\le d^{1-\frac{\pi}{4}}\int_{-\frac{\pi}{2}}^{\frac{\pi}{2}}\|\phi\|_\pi\, d\exp\left(kt^2\left|e^{\mathrm{i}\phi}\right|^2\left(1+\frac{8}{1-\lambda}\right)\right)d\mu(\phi)$$

$$= \|\phi\|_\pi\, d^{2-\frac{\pi}{4}}\exp\left(kt^2\left(1+\frac{8}{1-\lambda}\right)\right)\int_{-\frac{\pi}{2}}^{\frac{\pi}{2}}d\mu(\phi)$$

$$= \|\phi\|_\pi\, d^{2-\frac{\pi}{4}}\exp\left(kt^2\left(1+\frac{8}{1-\lambda}\right)\right)$$

where the first step follows Equation 4, the second step follows by swapping $\mathbb{E}$ and $\int$, the third step follows by Lemma 1, the forth step follows by $|e^{\mathrm{i}\phi}| = 1$, and the last step follows by $\mu$ is a probability distribution on $[-\frac{\pi}{2}, \frac{\pi}{2}]$ so $\int_{-\frac{\pi}{2}}^{\frac{\pi}{2}}d\mu(\phi) = 1$

Finally, putting it all together:

$$\mathbb{P}\left[\lambda_{\max}\left(\frac{1}{k}\sum_{j=1}^{k}f(v_j)\right) \ge \epsilon\right] \le \frac{\mathbb{E}\left[\mathrm{Tr}\left[\exp\left(t\sum_{j=1}^{k}f(v_j)\right)\right]\right]}{\exp(tk\epsilon)}$$

$$= \frac{\mathbb{E}\left[\mathrm{Tr}\left[\exp\left(\frac{\pi}{4}\left(\frac{4}{\pi}t\right)\sum_{j=1}^{k}f(v_j)\right)\right]\right]}{\exp(tk\epsilon)}$$

$$\le \frac{\|\phi\|_\pi\, d^{2-\frac{\pi}{4}}\exp\left(k\left(\frac{4}{\pi}t\right)^2\left(1+\frac{8}{1-\lambda}\right)\right)}{\exp(tk\epsilon)}$$

$$= \|\phi\|_\pi\, d^{2-\frac{\pi}{4}}\exp\left(\left(\frac{4}{\pi}\right)^2k\epsilon^2(1-\lambda)^2\frac{1}{36^2}\frac{9}{1-\lambda} - k\frac{(1-\lambda)\epsilon}{36}\epsilon\right)$$

$$\le \|\phi\|_\pi\, d^2\exp\left(-k\epsilon^2(1-\lambda)/72\right).$$

where the first step follows by Equation 3, the second step follows by Equation 5, the third step follows by choosing $t = (1-\lambda)\epsilon/36$. The only thing to be check is that $t = (1-\lambda)\epsilon/36$ satisfies $t\sqrt{\gamma^2 + b^2} = t \le \frac{1-\lambda}{4\lambda}$. Recall that $\epsilon < 1$ and $\lambda \le 1$, we have $t = \frac{(1-\lambda)\epsilon}{36} \le \frac{1-\lambda}{4} \le \frac{1-\lambda}{4\lambda}$. $\qquad\square$

## B.3 Proof of Lemma 1

**Lemma 1** (Analogous to Lemma 4.3 in [10]). *Let $\boldsymbol{P}$ be a regular Markov chain with state space $[N]$ with spectral expansion $\lambda$. Let $f$ be a function $f : [N] \to \mathbb{R}^{d \times d}$ such that (1) $\sum_{v \in [N]} \pi_v f(v) = 0$; (2) $\|f(v)\|_2 \leq 1$ and $f(v)$ is symmetric, $v \in [N]$. Let $(v_1, \cdots, v_k)$ denote a $k$-step random walk on $\boldsymbol{P}$ starting from a distribution $\phi$ on $[N]$. Then for any $t > 0, \gamma \geq 0, b > 0$ such that $t^2(\gamma^2 + b^2) \leq 1$ and $t\sqrt{\gamma^2 + b^2} \leq \frac{1-\lambda}{4\lambda}$, we have*

$$\mathbb{E}\left[\operatorname{Tr}\left[\prod_{j=1}^{k} \exp\left(\frac{tf(v_j)(\gamma + ib)}{2}\right) \prod_{j=k}^{1} \exp\left(\frac{tf(v_j)(\gamma - ib)}{2}\right)\right]\right] \leq \|\phi\|_{\pi} \, d \exp\left(kt^2(\gamma^2 + b^2)\left(1 + \frac{8}{1-\lambda}\right)\right).$$

*Proof.* Note that for $\boldsymbol{A}, \boldsymbol{B} \in \mathbb{C}^{d \times d}$, $\langle (\boldsymbol{A} \otimes \boldsymbol{B}) \operatorname{vec}(\boldsymbol{I}_d), \operatorname{vec}(\boldsymbol{I}_d) \rangle = \operatorname{Tr}[\boldsymbol{A}\boldsymbol{B}^\top]$. By letting $\boldsymbol{A} = \prod_{j=1}^{k} \exp\left(\frac{tf(v_j)(\gamma + ib)}{2}\right)$ and $\boldsymbol{B} = \left(\prod_{j=k}^{1} \exp\left(\frac{tf(v_j)(\gamma - ib)}{2}\right)\right)^\top = \prod_{j=1}^{k} \exp\left(\frac{tf(v_j)(\gamma - ib)}{2}\right)$. The trace term in LHS of Lemma 1 becomes

$$\operatorname{Tr}\left[\prod_{j=1}^{k} \exp\left(\frac{tf(v_j)(\gamma + ib)}{2}\right) \prod_{j=k}^{1} \exp\left(\frac{tf(v_j)(\gamma - ib)}{2}\right)\right]$$
$$= \left\langle \left(\prod_{j=1}^{k} \exp\left(\frac{tf(v_j)(\gamma + ib)}{2}\right) \otimes \prod_{j=1}^{k} \exp\left(\frac{tf(v_j)(\gamma - ib)}{2}\right)\right) \operatorname{vec}(\boldsymbol{I}_d), \operatorname{vec}(\boldsymbol{I}_d) \right\rangle. \tag{6}$$

By iteratively applying $(\boldsymbol{A} \otimes \boldsymbol{B})(\boldsymbol{C} \otimes \boldsymbol{D}) = (\boldsymbol{A}\boldsymbol{C}) \otimes (\boldsymbol{B}\boldsymbol{D})$, we have

$$\prod_{j=1}^{k} \exp\left(\frac{tf(v_j)(\gamma + ib)}{2}\right) \otimes \prod_{j=1}^{k} \exp\left(\frac{tf(v_j)(\gamma - ib)}{2}\right)$$
$$= \prod_{j=1}^{k} \left(\exp\left(\frac{tf(v_j)(\gamma + ib)}{2}\right) \otimes \exp\left(\frac{tf(v_j)(\gamma - ib)}{2}\right)\right) \triangleq \prod_{j=1}^{k} \boldsymbol{M}_{v_j},$$

where we define

$$\boldsymbol{M}_{v_j} \triangleq \exp\left(\frac{tf(v_j)(\gamma + ib)}{2}\right) \otimes \exp\left(\frac{tf(v_j)(\gamma - ib)}{2}\right). \tag{7}$$

Plug it to the trace term, we have

$$\operatorname{Tr}\left[\prod_{j=1}^{k} \exp\left(\frac{tf(v_j)(\gamma + ib)}{2}\right) \prod_{j=k}^{1} \exp\left(\frac{tf(v_j)(\gamma - ib)}{2}\right)\right] = \left\langle \left(\prod_{j=1}^{k} \boldsymbol{M}_{v_j}\right) \operatorname{vec}(\boldsymbol{I}_d), \operatorname{vec}(\boldsymbol{I}_d) \right\rangle.$$

Next, taking expectation on Equation 6 gives

$$\mathbb{E}_{v_1, \cdots, v_k}\left[\operatorname{Tr}\left[\prod_{j=1}^{k} \exp\left(\frac{tf(v_j)(\gamma + ib)}{2}\right) \prod_{j=k}^{1} \exp\left(\frac{tf(v_j)(\gamma - ib)}{2}\right)\right]\right]$$
$$= \mathbb{E}_{v_1, \cdots, v_k}\left[\left\langle \left(\prod_{j=1}^{k} \boldsymbol{M}_{v_j}\right) \operatorname{vec}(\boldsymbol{I}_d), \operatorname{vec}(\boldsymbol{I}_d) \right\rangle\right] \tag{8}$$
$$= \left\langle \mathbb{E}_{v_1, \cdots, v_k}\left[\prod_{j=1}^{k} \boldsymbol{M}_{v_j}\right] \operatorname{vec}(\boldsymbol{I}_d), \operatorname{vec}(\boldsymbol{I}_d) \right\rangle.$$

We turn to study $\mathbb{E}_{v_1, \cdots, v_k}\left[\prod_{j=1}^{k} \boldsymbol{M}_{v_j}\right]$, which is characterized by the following lemma:

**Lemma 2.** *Let $\boldsymbol{E} \triangleq \operatorname{diag}(\boldsymbol{M}_1, \boldsymbol{M}_2, \cdots, \boldsymbol{M}_N) \in \mathbb{C}^{Nd^2 \times Nd^2}$ and $\widetilde{\boldsymbol{P}} \triangleq \boldsymbol{P} \otimes \boldsymbol{I}_{d^2} \in \mathbb{R}^{Nd^2 \times Nd^2}$. For a random walk $(v_1, \cdots, v_k)$ such that $v_1$ is sampled from an arbitrary probability distribution $\phi$ on $[N]$, $\mathbb{E}_{v_1, \cdots, v_k}\left[\prod_{j=1}^{k} \boldsymbol{M}_{v_j}\right] = (\phi \otimes \boldsymbol{I}_{d^2})^\top \left((\boldsymbol{E}\widetilde{\boldsymbol{P}})^{k-1}\boldsymbol{E}\right)(\mathbf{1} \otimes \boldsymbol{I}_{d^2})$, where $\mathbf{1}$ is the all-ones vector.*

*Proof.* (of Lemma 2) We always treat $\boldsymbol{E}\widetilde{\boldsymbol{P}}$ as a block matrix, s.t.,

$$\boldsymbol{E}\widetilde{\boldsymbol{P}} = \begin{bmatrix} \boldsymbol{M}_1 & & \\ & \ddots & \\ & & \boldsymbol{M}_N \end{bmatrix} \begin{bmatrix} \boldsymbol{P}_{1,1}\boldsymbol{I}_{d^2} & \cdots & \boldsymbol{P}_{1,N}\boldsymbol{I}_{d^2} \\ \vdots & \ddots & \vdots \\ \boldsymbol{P}_{N,1}\boldsymbol{I}_{d^2} & \cdots & \boldsymbol{P}_{N,N}\boldsymbol{I}_{d^2} \end{bmatrix} = \begin{bmatrix} \boldsymbol{P}_{1,1}\boldsymbol{M}_1 & \cdots & \boldsymbol{P}_{1,N}\boldsymbol{M}_1 \\ \vdots & \ddots & \vdots \\ \boldsymbol{P}_{N,1}\boldsymbol{M}_N & \cdots & \boldsymbol{P}_{N,N}\boldsymbol{M}_N \end{bmatrix}.$$

I.e., the $(u, v)$-th block of $\boldsymbol{E}\widetilde{\boldsymbol{P}}$, denoted by $(\boldsymbol{E}\widetilde{\boldsymbol{P}})_{u,v}$, is $\boldsymbol{P}_{u,v}\boldsymbol{M}_u$.

$$
\begin{aligned}
\mathbb{E}_{v_1,\cdots,v_k}\left[\prod_{j=1}^{k}\boldsymbol{M}_{v_j}\right] &= \sum_{v_1,\cdots,v_k}\boldsymbol{\phi}_{v_1}\boldsymbol{P}_{v_1,v_2}\cdots\boldsymbol{P}_{v_{k-1},v_k}\prod_{j=1}^{k}\boldsymbol{M}_{v_j}\\
&= \sum_{v_1}\boldsymbol{\phi}_{v_1}\sum_{v_2}\left(\boldsymbol{P}_{v_1,v_2}\boldsymbol{M}_{v_1}\right)\cdots\sum_{v_k}\left(\boldsymbol{P}_{v_{k-1},v_k}\boldsymbol{M}_{v_{k-1}}\right)\boldsymbol{M}_{v_k}\\
&= \sum_{v_1}\boldsymbol{\phi}_{v_1}\sum_{v_2}(\boldsymbol{E}\widetilde{\boldsymbol{P}})_{v_1,v_2}\sum_{v_3}(\boldsymbol{E}\widetilde{\boldsymbol{P}})_{v_2,v_3}\cdots\sum_{v_k}(\boldsymbol{E}\widetilde{\boldsymbol{P}}\boldsymbol{E})_{v_{k-1},v_k}\\
&= \sum_{v_1}\boldsymbol{\phi}_{v_1}\sum_{v_k}\left((\boldsymbol{E}\widetilde{\boldsymbol{P}})^{k-1}\boldsymbol{E}\right)_{v_1,v_k} = (\boldsymbol{\phi}\otimes\boldsymbol{I}_{d^2})^{\top}\left((\boldsymbol{E}\widetilde{\boldsymbol{P}})^{k-1}\boldsymbol{E}\right)(\boldsymbol{1}\otimes\boldsymbol{I}_{d^2})
\end{aligned}
$$

$\square$

Given Lemma 2, Equation 8 becomes:

$$
\mathbb{E}_{v_1,\cdots,v_k}\left[\mathrm{Tr}\left[\prod_{j=1}^{k}\exp\left(\frac{tf(v_j)(\gamma+\mathrm{i}b)}{2}\right)\prod_{j=k}^{1}\exp\left(\frac{tf(v_j)(\gamma-\mathrm{i}b)}{2}\right)\right]\right]
$$

$$
= \left\langle\mathbb{E}_{v_1,\cdots,v_k}\left[\prod_{j=1}^{k}\boldsymbol{M}_{v_j}\right]\mathrm{vec}(\boldsymbol{I}_d),\mathrm{vec}(\boldsymbol{I}_d)\right\rangle
$$

$$
= \left\langle(\boldsymbol{\phi}\otimes\boldsymbol{I}_{d^2})^{\top}\left((\boldsymbol{E}\widetilde{\boldsymbol{P}})^{k-1}\boldsymbol{E}\right)(\boldsymbol{1}\otimes\boldsymbol{I}_{d^2}),\mathrm{vec}(\boldsymbol{I}_d)\right\rangle
$$

$$
= \left\langle\left((\boldsymbol{E}\widetilde{\boldsymbol{P}})^{k-1}\boldsymbol{E}\right)(\boldsymbol{1}\otimes\boldsymbol{I}_{d^2})\mathrm{vec}(\boldsymbol{I}_d),(\boldsymbol{\phi}\otimes\boldsymbol{I}_{d^2})\mathrm{vec}(\boldsymbol{I}_d)\right\rangle
$$

$$
= \left\langle\left((\boldsymbol{E}\widetilde{\boldsymbol{P}})^{k-1}\boldsymbol{E}\right)(\boldsymbol{1}\otimes\mathrm{vec}(\boldsymbol{I}_d)),\boldsymbol{\pi}\otimes\mathrm{vec}(\boldsymbol{I}_d)\right\rangle
$$

The third equality is due to $\langle x, \boldsymbol{A}y\rangle = \langle \boldsymbol{A}^*x, y\rangle$. The forth equality is by setting $\boldsymbol{C} = 1$ (scalar) in $(\boldsymbol{A}\otimes\boldsymbol{B})(\boldsymbol{C}\otimes\boldsymbol{D}) = (\boldsymbol{A}\boldsymbol{C})\otimes(\boldsymbol{B}\boldsymbol{D})$. Then

$$
\mathbb{E}_{v_1,\cdots,v_k}\left[\mathrm{Tr}\left[\prod_{j=1}^{k}\exp\left(\frac{tf(v_j)(\gamma+\mathrm{i}b)}{2}\right)\prod_{j=k}^{1}\exp\left(\frac{tf(v_j)(\gamma-\mathrm{i}b)}{2}\right)\right]\right]
$$

$$
= \left\langle\left((\boldsymbol{E}\widetilde{\boldsymbol{P}})^{k-1}\boldsymbol{E}\right)(\boldsymbol{1}\otimes\mathrm{vec}(\boldsymbol{I}_d)),\boldsymbol{\phi}\otimes\mathrm{vec}(\boldsymbol{I}_d)\right\rangle
$$

$$
= (\boldsymbol{\phi}\otimes\mathrm{vec}(\boldsymbol{I}_d))^*\left((\boldsymbol{E}\widetilde{\boldsymbol{P}})^{k-1}\boldsymbol{E}\right)(\boldsymbol{1}\otimes\mathrm{vec}(\boldsymbol{I}_d))
$$

$$
= (\boldsymbol{\phi}\otimes\mathrm{vec}(\boldsymbol{I}_d))^*\left((\boldsymbol{E}\widetilde{\boldsymbol{P}})^{k-1}\boldsymbol{E}\right)\left((\boldsymbol{P}\boldsymbol{\Pi}^{-1}\boldsymbol{\pi})\otimes(\boldsymbol{I}_{d^2}\boldsymbol{I}_{d^2}\mathrm{vec}(\boldsymbol{I}_d))\right)
$$

$$
= (\boldsymbol{\phi}\otimes\mathrm{vec}(\boldsymbol{I}_d))^*\left(\boldsymbol{E}\widetilde{\boldsymbol{P}}\right)^k(\boldsymbol{\Pi}^{-1}\otimes\boldsymbol{I}_{d^2})(\boldsymbol{\pi}\otimes\mathrm{vec}(\boldsymbol{I}_d)) \triangleq \langle\boldsymbol{\pi}\otimes\mathrm{vec}(\boldsymbol{I}_d),\boldsymbol{z}_k\rangle_{\boldsymbol{\pi}},
$$

where we define $\boldsymbol{z}_0 = \boldsymbol{\phi}\otimes\mathrm{vec}(\boldsymbol{I}_d)$ and $\boldsymbol{z}_k = \left(\boldsymbol{z}_0^*\left(\boldsymbol{E}\widetilde{\boldsymbol{P}}\right)^k\right)^* = \left(\boldsymbol{z}_{k-1}^*\boldsymbol{E}\widetilde{\boldsymbol{P}}\right)^*$. Moreover, by Remark 2, we have $\|\boldsymbol{\pi}\otimes\mathrm{vec}(\boldsymbol{I}_d)\|_{\boldsymbol{\pi}} = \|\boldsymbol{\pi}\|_{\boldsymbol{\pi}}\|\mathrm{vec}(\boldsymbol{I}_d)\|_2 = \sqrt{d}$ and $\|\boldsymbol{z}_0\|_{\boldsymbol{\pi}} = \|\boldsymbol{\phi}\otimes\mathrm{vec}(\boldsymbol{I}_d)\|_{\boldsymbol{\pi}} = \|\boldsymbol{\phi}\|_{\boldsymbol{\pi}}\|\mathrm{vec}(\boldsymbol{I}_d)\|_2 = \|\boldsymbol{\phi}\|_{\boldsymbol{\pi}}\sqrt{d}$

**Definition 2.** *Define linear subspace* $\mathcal{U} = \left\{\boldsymbol{\pi}\otimes\boldsymbol{w}, \boldsymbol{w}\in\mathbb{C}^{d^2}\right\}$.

**Remark 3.** $\{\boldsymbol{\pi}\otimes\boldsymbol{e}_i, i\in[d^2]\}$ *is an orthonormal basis of* $\mathcal{U}$. *This is because* $\langle\boldsymbol{\pi}\otimes\boldsymbol{e}_i,\boldsymbol{\pi}\otimes\boldsymbol{e}_j\rangle_{\boldsymbol{\pi}} = \langle\boldsymbol{\pi},\boldsymbol{\pi}\rangle_{\boldsymbol{\pi}}\langle\boldsymbol{e}_i,\boldsymbol{e}_j\rangle = \delta_{ij}$ *by Remark 1, where* $\delta_{ij}$ *is the Kronecker delta.*

**Remark 4.** *Given* $\boldsymbol{x} = \boldsymbol{y}\otimes\boldsymbol{w}$. *The projection of* $\boldsymbol{x}$ *on to* $\mathcal{U}$ *is* $\boldsymbol{x}^{\|} = (\boldsymbol{1}^*\boldsymbol{y})(\boldsymbol{\pi}\otimes\boldsymbol{w})$. *This is because*

$$
\boldsymbol{x}^{\|} = \sum_{i=1}^{d^2}\langle\boldsymbol{y}\otimes\boldsymbol{w},\boldsymbol{\pi}\otimes\boldsymbol{e}_i\rangle_{\boldsymbol{\pi}}(\boldsymbol{\pi}\otimes\boldsymbol{e}_i) = \sum_{i=1}^{d^2}\langle\boldsymbol{y},\boldsymbol{\pi}\rangle_{\boldsymbol{\pi}}\langle\boldsymbol{w},\boldsymbol{e}_i\rangle(\boldsymbol{\pi}\otimes\boldsymbol{e}_i) = (\boldsymbol{1}^*\boldsymbol{y})(\boldsymbol{\pi}\otimes\boldsymbol{w}).
$$

We want to bound

$$
\langle\boldsymbol{\pi}\otimes\mathrm{vec}(\boldsymbol{I}_d),\boldsymbol{z}_k\rangle_{\boldsymbol{\pi}} = \left\langle\boldsymbol{\pi}\otimes\mathrm{vec}(\boldsymbol{I}_d),\boldsymbol{z}_k^{\perp}+\boldsymbol{z}_k^{\|}\right\rangle_{\boldsymbol{\pi}} = \left\langle\boldsymbol{\pi}\otimes\mathrm{vec}(\boldsymbol{I}_d),\boldsymbol{z}_k^{\|}\right\rangle_{\boldsymbol{\pi}}
$$

$$
\leq \|\boldsymbol{\pi}\otimes\mathrm{vec}(\boldsymbol{I}_d)\|_{\boldsymbol{\pi}}\left\|\boldsymbol{z}_k^{\|}\right\|_{\boldsymbol{\pi}} = \sqrt{d}\left\|\boldsymbol{z}_k^{\|}\right\|_{\boldsymbol{\pi}}.
$$

As $\boldsymbol{z}_k$ can be expressed as recursively applying operator $\boldsymbol{E}$ and $\widetilde{\boldsymbol{P}}$ on $\boldsymbol{z}_0$, we turn to analyze the effects of $\boldsymbol{E}$ and $\widetilde{\boldsymbol{P}}$ operators.

**Definition 3.** *The spectral expansion of $\widetilde{\boldsymbol{P}}$ is defined as $\lambda(\widetilde{\boldsymbol{P}}) \triangleq \max_{\boldsymbol{x} \perp \mathcal{U}, \boldsymbol{x} \neq 0} \frac{\left\|\left(\boldsymbol{x}^* \widetilde{\boldsymbol{P}}\right)^*\right\|_{\boldsymbol{\pi}}}{\|\boldsymbol{x}\|_{\boldsymbol{\pi}}}$*

**Lemma 3.** $\lambda(\boldsymbol{P}) = \lambda(\widetilde{\boldsymbol{P}})$.

*Proof.* First show $\lambda(\widetilde{\boldsymbol{P}}) \geq \lambda(\boldsymbol{P})$. Suppose the maximizer of $\lambda(\boldsymbol{P}) \triangleq \max_{\boldsymbol{y} \perp \boldsymbol{\pi}, \boldsymbol{y} \neq 0} \frac{\|(\boldsymbol{y}^* \boldsymbol{P})^*\|_{\boldsymbol{\pi}}}{\|\boldsymbol{y}\|_{\boldsymbol{\pi}}}$ is $\boldsymbol{y} \in \mathbb{C}^n$, i.e., $\left\|(\boldsymbol{y}^* \boldsymbol{P})^*\right\|_{\boldsymbol{\pi}} = \lambda(\boldsymbol{P}) \|\boldsymbol{y}\|_{\boldsymbol{\pi}}$. Construct $\boldsymbol{x} = \boldsymbol{y} \otimes \boldsymbol{o}$ for arbitrary non-zero $\boldsymbol{o} \in \mathbb{C}^{d^2}$. Easy to check that $\boldsymbol{x} \perp \mathcal{U}$, because $\langle \boldsymbol{x}, \boldsymbol{\pi} \otimes \boldsymbol{w} \rangle_{\boldsymbol{\pi}} = \langle \boldsymbol{y}, \boldsymbol{\pi} \rangle_{\boldsymbol{\pi}} \langle \boldsymbol{o}, \boldsymbol{w} \rangle = 0$, where the last equality is due to $\boldsymbol{y} \perp \boldsymbol{\pi}$. Then we can bound $\left\|\left(\boldsymbol{x}^* \widetilde{\boldsymbol{P}}\right)^*\right\|_{\boldsymbol{\pi}}$ such that

$$\left\|\left(\boldsymbol{x}^* \widetilde{\boldsymbol{P}}\right)^*\right\|_{\boldsymbol{\pi}} = \left\|\widetilde{\boldsymbol{P}}^* \boldsymbol{x}\right\|_{\boldsymbol{\pi}} = \|(\boldsymbol{P}^* \otimes \boldsymbol{I}_{d^2})(\boldsymbol{y} \otimes \boldsymbol{o})\|_{\boldsymbol{\pi}} = \|(\boldsymbol{P}^* \boldsymbol{y}) \otimes \boldsymbol{o}\|_{\boldsymbol{\pi}}$$
$$= \|(\boldsymbol{y}^* \boldsymbol{P})^*\|_{\boldsymbol{\pi}} \|\boldsymbol{o}\|_2 = \lambda(\boldsymbol{P}) \|\boldsymbol{y}\|_{\boldsymbol{\pi}} \|\boldsymbol{o}\|_2 = \lambda(\boldsymbol{P}) \|\boldsymbol{x}\|_{\boldsymbol{\pi}},$$

which indicate for $\boldsymbol{x} = \boldsymbol{y} \otimes \boldsymbol{o}$, $\frac{\|(\boldsymbol{x}^* \widetilde{\boldsymbol{P}})^*\|_{\boldsymbol{\pi}}}{\|\boldsymbol{x}\|_{\boldsymbol{\pi}}} = \lambda(\boldsymbol{P})$. Taking maximum over all $\boldsymbol{x}$ gives $\lambda(\widetilde{\boldsymbol{P}}) \geq \lambda(\boldsymbol{P})$.

Next to show $\lambda(\boldsymbol{P}) \geq \lambda(\widetilde{\boldsymbol{P}})$. For $\forall \boldsymbol{x} \in \mathbb{C}^{Nd^2}$ such that $\boldsymbol{x} \perp \mathcal{U}$ and $\boldsymbol{x} \neq 0$, we can decompose it to be

$$\boldsymbol{x} = \begin{bmatrix} x_1 \\ x_2 \\ \vdots \\ x_{Nd^2} \end{bmatrix} = \begin{bmatrix} x_1 \\ x_{d^2+1} \\ \vdots \\ x_{(N-1)d^2+1} \end{bmatrix} \otimes \boldsymbol{e}_1 + \begin{bmatrix} x_2 \\ x_{d^2+2} \\ \vdots \\ x_{(N-1)d^2+2} \end{bmatrix} \otimes \boldsymbol{e}_2 + \cdots + \begin{bmatrix} x_{d^2} \\ x_{2d^2} \\ \vdots \\ x_{Nd^2} \end{bmatrix} \otimes \boldsymbol{e}_{d^2} \triangleq \sum_{i=1}^{d^2} \boldsymbol{x}_i \otimes \boldsymbol{e}_i,$$

where we define $\boldsymbol{x}_i \triangleq \begin{bmatrix} x_i & \cdots & x_{(N-1)d^2+i} \end{bmatrix}^\top$ for $i \in [d^2]$. We can observe that $\boldsymbol{x}_i \perp \boldsymbol{\pi}, i \in [d^2]$, because for $\forall j \in [d^2]$, we have

$$0 = \langle \boldsymbol{x}, \boldsymbol{\pi} \otimes \boldsymbol{e}_j \rangle_{\boldsymbol{\pi}} = \left\langle \sum_{i=1}^{d^2} \boldsymbol{x}_i \otimes \boldsymbol{e}_i, \boldsymbol{\pi} \otimes \boldsymbol{e}_j \right\rangle_{\boldsymbol{\pi}} = \sum_{i=1}^{d^2} \langle \boldsymbol{x}_i \otimes \boldsymbol{e}_i, \boldsymbol{\pi} \otimes \boldsymbol{e}_j \rangle_{\boldsymbol{\pi}} = \sum_{i=1}^{d^2} \langle \boldsymbol{x}_i, \boldsymbol{\pi} \rangle_{\boldsymbol{\pi}} \langle \boldsymbol{e}_i, \boldsymbol{e}_j \rangle = \langle \boldsymbol{x}_j, \boldsymbol{\pi} \rangle_{\boldsymbol{\pi}},$$

which indicates $\boldsymbol{x}_j \perp \boldsymbol{\pi}, j \in [d^2]$. Furthermore, we can also observe that $\boldsymbol{x}_i \otimes \boldsymbol{e}_i, i \in [d^2]$ is pairwise orthogonal. This is because for $\forall i, j \in [d^2]$, $\langle \boldsymbol{x}_i \otimes \boldsymbol{e}_i, \boldsymbol{x}_j \otimes \boldsymbol{e}_j \rangle_{\boldsymbol{\pi}} = \langle \boldsymbol{x}_i, \boldsymbol{x}_j \rangle_{\boldsymbol{\pi}} \langle \boldsymbol{e}_i, \boldsymbol{e}_j \rangle = \delta_{ij}$, which suggests us to use Pythagorean theorem such that $\|\boldsymbol{x}\|_{\boldsymbol{\pi}}^2 = \sum_{i=1}^{d^2} \|\boldsymbol{x}_i \otimes \boldsymbol{e}_i\|_{\boldsymbol{\pi}}^2 = \sum_{i=1}^{d^2} \|\boldsymbol{x}_i\|_{\boldsymbol{\pi}} \|\boldsymbol{e}_i\|_2^2$.

We can use similar way to decompose and analyze $\left(\boldsymbol{x}^* \widetilde{\boldsymbol{P}}\right)^*$:

$$\left(\boldsymbol{x}^* \widetilde{\boldsymbol{P}}\right)^* = \widetilde{\boldsymbol{P}}^* \boldsymbol{x} = \sum_{i=1}^{d^2} (\boldsymbol{P}^* \otimes \boldsymbol{I}_{d^2})(\boldsymbol{x}_i \otimes \boldsymbol{e}_i) = \sum_{i=1}^{d^2} (\boldsymbol{P}^* \boldsymbol{x}_i) \otimes \boldsymbol{e}_i.$$

where we can observe that $(\boldsymbol{P}^* \boldsymbol{x}_i) \otimes \boldsymbol{e}_i, i \in [d^2]$ is pairwise orthogonal. This is because for $\forall i, j \in [d^2]$, we have $\langle (\boldsymbol{P}^* \boldsymbol{x}_i) \otimes \boldsymbol{e}_i, (\boldsymbol{P}^* \boldsymbol{x}_j) \otimes \boldsymbol{e}_j \rangle_{\boldsymbol{\pi}} = \langle \boldsymbol{P}^* \boldsymbol{x}_i, \boldsymbol{P}^* \boldsymbol{x}_j \rangle_{\boldsymbol{\pi}} \langle \boldsymbol{e}_i, \boldsymbol{e}_j \rangle = \delta_{ij}$. Again, applying Pythagorean theorem gives:

$$\left\|\left(\boldsymbol{x}^* \widetilde{\boldsymbol{P}}\right)^*\right\|_{\boldsymbol{\pi}}^2 = \sum_{i=1}^{d^2} \|(\boldsymbol{P}^* \boldsymbol{x}_i) \otimes \boldsymbol{e}_i\|_{\boldsymbol{\pi}}^2 = \sum_{i=1}^{d^2} \|(\boldsymbol{x}_i^* \boldsymbol{P})^*\|_{\boldsymbol{\pi}}^2 \|\boldsymbol{e}_i\|_2^2$$
$$\leq \sum_{i=1}^{d^2} \lambda(\boldsymbol{P})^2 \|\boldsymbol{x}_i\|_{\boldsymbol{\pi}}^2 \|\boldsymbol{e}_i\|_2^2 = \lambda(\boldsymbol{P})^2 \left( \sum_{i=1}^{d^2} \|\boldsymbol{x}_i\|_{\boldsymbol{\pi}}^2 \|\boldsymbol{e}_i\|_2^2 \right) = \lambda(\boldsymbol{P})^2 \|\boldsymbol{x}\|_{\boldsymbol{\pi}}^2,$$

which indicate that for $\forall \boldsymbol{x}$ such that $\boldsymbol{x} \perp \mathcal{U}$ and $\boldsymbol{x} \neq 0$, we have $\frac{\|(\boldsymbol{x}^* \widetilde{\boldsymbol{P}})^*\|_{\boldsymbol{\pi}}}{\|\boldsymbol{x}\|_{\boldsymbol{\pi}}} \leq \lambda(\boldsymbol{P})$, or equivalently $\lambda(\widetilde{\boldsymbol{P}}) \leq \lambda(\boldsymbol{P})$.

Overall, we have shown both $\lambda(\widetilde{\boldsymbol{P}}) \geq \lambda(\boldsymbol{P})$ and $\lambda(\widetilde{\boldsymbol{P}}) \leq \lambda(\boldsymbol{P})$. We conclude $\lambda(\widetilde{\boldsymbol{P}}) = \lambda(\boldsymbol{P})$. $\quad \square$

**Lemma 4.** *(**The effect of $\widetilde{P}$ operator**) This lemma is a generalization of lemma 3.3 in [6].*

1. $\forall y \in \mathcal{U}$, then $\left(y^* \widetilde{P}\right)^* = y$.

2. $\forall y \perp \mathcal{U}$, then $\left(y^* \widetilde{P}\right)^* \perp \mathcal{U}$, and $\left\|\left(y^* \widetilde{P}\right)^*\right\|_{\boldsymbol{\pi}} \leq \lambda \left\|y\right\|_{\boldsymbol{\pi}}$.

*Proof.* First prove the Part 1 of lemma 4. $\forall y = \boldsymbol{\pi} \otimes \boldsymbol{w} \in \mathcal{U}$:

$$y^* \widetilde{P} = (\boldsymbol{\pi}^* \otimes \boldsymbol{w}^*)\left(\boldsymbol{P} \otimes \boldsymbol{I}_{d^2}\right) = (\boldsymbol{\pi}^* \boldsymbol{P}) \otimes (\boldsymbol{w}^* \boldsymbol{I}_{d^2}) = \boldsymbol{\pi}^* \otimes \boldsymbol{w}^* = \boldsymbol{y}^*,$$

where third equality is becase $\boldsymbol{\pi}$ is the stationary distribution. Next to prove Part 2 of lemma 4. Given $y \perp \mathcal{U}$, want to show $(y^* \widetilde{P})^* \perp \boldsymbol{\pi} \otimes \boldsymbol{w}$, for every $\boldsymbol{w} \in \mathbb{C}^{d^2}$. It is true because

$$\left\langle \boldsymbol{\pi} \otimes \boldsymbol{w}, (y^* \widetilde{P})^* \right\rangle_{\boldsymbol{\pi}} = y^* \widetilde{P}\left(\boldsymbol{\Pi}^{-1} \otimes \boldsymbol{I}_{d^2}\right)(\boldsymbol{\pi} \otimes \boldsymbol{w}) = y^*\left((\boldsymbol{P}\boldsymbol{\Pi}^{-1}\boldsymbol{\pi}) \otimes \boldsymbol{w}\right) = y^*\left((\boldsymbol{\Pi}^{-1}\boldsymbol{\pi}) \otimes \boldsymbol{w}\right)$$
$$= y^*\left(\boldsymbol{\Pi}^{-1} \otimes \boldsymbol{I}_{d^2}\right)(\boldsymbol{\pi} \otimes \boldsymbol{w}) = \langle \boldsymbol{\pi} \otimes \boldsymbol{w}, y \rangle_{\boldsymbol{\pi}} = 0.$$

The third equality is due to $\boldsymbol{P}\boldsymbol{\Pi}^{-1}\boldsymbol{\pi} = \boldsymbol{P}\boldsymbol{1} = \boldsymbol{1} = \boldsymbol{\Pi}^{-1}\boldsymbol{\pi}$. Moreover, $\left\|\left(y^* \widetilde{P}\right)^*\right\|_{\boldsymbol{\pi}} \leq \lambda \left\|y\right\|_{\boldsymbol{\pi}}$ is simply a re-statement of definition 3. $\qquad\square$

**Remark 5.** *Lemma 4 implies that $\forall y \in \mathbb{C}^{nd^2}$*

1. $\left(\left(y^* \widetilde{P}\right)^*\right)^{\|} = \left(\left(y^{\|*} \widetilde{P}\right)^*\right)^{\|} + \left(\left(y^{\perp*} \widetilde{P}\right)^*\right)^{\|} = y^{\|} + 0 = y^{\|}$

2. $\left(\left(y^* \widetilde{P}\right)^*\right)^{\perp} = \left(\left(y^{\|*} \widetilde{P}\right)^*\right)^{\perp} + \left(\left(y^{\perp*} \widetilde{P}\right)^*\right)^{\perp} = 0 + \left(y^{\perp*} \widetilde{P}\right)^* = \left(y^{\perp*} \widetilde{P}\right)^*$.

**Lemma 5.** *(**The effect of $E$ operator**) Given three parameters $\lambda \in [0,1], \ell \geq 0$ and $t > 0$. Let $\boldsymbol{P}$ be a regular Markov chain on state space $[N]$, with stationary distribution $\boldsymbol{\pi}$ and spectral expansion $\lambda$. Suppose each state $i \in [N]$ is assigned a matrix $\boldsymbol{H}_i \in \mathbb{C}^{d^2 \times d^2}$ s.t. $\|\boldsymbol{H}_i\|_2 \leq \ell$ and $\sum_{i \in [N]} \pi_i \boldsymbol{H}_i = 0$. Let $\widetilde{P} = \boldsymbol{P} \otimes \boldsymbol{I}_{d^2}$ and $\boldsymbol{E}$ denotes the $Nd^2 \times Nd^2$ block matrix where the $i$-th diagonal block is the matrix $\exp(t\boldsymbol{H}_i)$, i.e., $\boldsymbol{E} = \mathrm{diag}(\exp(t\boldsymbol{H}_1), \cdots, \exp(t\boldsymbol{H}_N))$. Then for any $\forall z \in \mathbb{C}^{Nd^2}$, we have:*

1. $\left\|\left(\left(z^{\|*} \boldsymbol{E} \widetilde{P}\right)^*\right)^{\|}\right\|_{\boldsymbol{\pi}} \leq \alpha_1 \left\|z^{\|}\right\|_{\boldsymbol{\pi}}$, where $\alpha_1 = \exp(t\ell) - t\ell$.

2. $\left\|\left(\left(z^{\|*} \boldsymbol{E} \widetilde{P}\right)^*\right)^{\perp}\right\|_{\boldsymbol{\pi}} \leq \alpha_2 \left\|z^{\|}\right\|_{\boldsymbol{\pi}}$, where $\alpha_2 = \lambda(\exp(t\ell) - 1)$.

3. $\left\|\left(\left(z^{\perp*} \boldsymbol{E} \widetilde{P}\right)^*\right)^{\|}\right\|_{\boldsymbol{\pi}} \leq \alpha_3 \left\|z^{\perp}\right\|_{\boldsymbol{\pi}}$, where $\alpha_3 = \exp(t\ell) - 1$.

4. $\left\|\left(\left(z^{\perp*} \boldsymbol{E} \widetilde{P}\right)^*\right)^{\perp}\right\|_{\boldsymbol{\pi}} \leq \alpha_4 \left\|z^{\perp}\right\|_{\boldsymbol{\pi}}$, where $\alpha_4 = \lambda \exp(t\ell)$.

*Proof.* (of Lemma 5) We first show that, for $z = y \otimes \boldsymbol{w}$,

$$(z^* \boldsymbol{E})^* = \boldsymbol{E}^* z = \begin{bmatrix} \exp(t\boldsymbol{H}_1^*) & & \\ & \ddots & \\ & & \exp(t\boldsymbol{H}_N^*) \end{bmatrix} \begin{bmatrix} y_1 \boldsymbol{w} \\ \vdots \\ y_N \boldsymbol{w} \end{bmatrix} = \begin{bmatrix} y_1 \exp(t\boldsymbol{H}_1^*)\boldsymbol{w} \\ \vdots \\ y_N \exp(t\boldsymbol{H}_N^*)\boldsymbol{w} \end{bmatrix}$$

$$= \begin{bmatrix} y_1 \exp(t\boldsymbol{H}_1^*)\boldsymbol{w} \\ \vdots \\ 0 \end{bmatrix} + \cdots + \begin{bmatrix} 0 \\ \vdots \\ y_N \exp(t\boldsymbol{H}_N^*)\boldsymbol{w} \end{bmatrix} = \sum_{i=1}^{N} y_i \left(\boldsymbol{e}_i \otimes (\exp(t\boldsymbol{H}_i^*)\boldsymbol{w})\right).$$

Due to the linearity of projection,

$$\left((\boldsymbol{z}^*\boldsymbol{E})^*\right)^{\|} = \sum_{i=1}^N y_i\left(\boldsymbol{e}_i \otimes (\exp(t\boldsymbol{H}_i^*)\boldsymbol{w})\right)^{\|} = \sum_{i=1}^N y_i(\mathbf{1}^*\boldsymbol{e}_i)\left(\boldsymbol{\pi} \otimes (\exp(t\boldsymbol{H}_i^*)\boldsymbol{w})\right) = \boldsymbol{\pi} \otimes \left(\sum_{i=1}^N y_i\exp(t\boldsymbol{H}_i^*)\boldsymbol{w}\right),$$

(9)

where the second inequality follows by Remark 4.

**Proof of Lemma 5, Part 1** Firstly We can bound $\left\|\sum_{i=1}^N \pi_i\exp(t\boldsymbol{H}_i^*)\right\|_2$ by

$$\left\|\sum_{i=1}^N \pi_i\exp(t\boldsymbol{H}_i^*)\right\|_2 = \left\|\sum_{i=1}^N \pi_i\exp(t\boldsymbol{H}_i)\right\|_2 = \left\|\sum_{i=1}^N \pi_i\sum_{k=0}^{+\infty}\frac{t^j\boldsymbol{H}_i^j}{j!}\right\|_2 = \left\|\boldsymbol{I} + \sum_{i=1}^N \pi_i\sum_{j=2}^{+\infty}\frac{t^j\boldsymbol{H}_i^j}{j!}\right\|_2$$

$$\leq 1 + \sum_{i=1}^N \pi_i\sum_{j=2}^{+\infty}\frac{t^j\|\boldsymbol{H}_i\|_2^j}{j!} \leq 1 + \sum_{i=1}^N \pi_i\sum_{j=2}^{+\infty}\frac{(t\ell)^j}{j!} = \exp(t\ell) - t\ell,$$

where the first step follows by $\|\boldsymbol{A}\|_2 = \|\boldsymbol{A}^*\|_2$, the second step follows by matrix exponential, the third step follows by $\sum_{i\in[N]}\pi_i\boldsymbol{H}_i = 0$, and the forth step follows by triangle inequality. Given the above bound, for any $\boldsymbol{z}^{\|}$ which can be written as $\boldsymbol{z}^{\|} = \boldsymbol{\pi} \otimes \boldsymbol{w}$ for some $\boldsymbol{w} \in \mathbb{C}^{d^2}$, we have

$$\left\|\left(\left(\boldsymbol{z}^{\|*}\boldsymbol{E}\widetilde{\boldsymbol{P}}\right)^*\right)^{\|}\right\|_{\boldsymbol{\pi}} = \left\|\left(\left(\boldsymbol{z}^{\|*}\boldsymbol{E}\right)^*\right)^{\|}\right\|_{\boldsymbol{\pi}} = \left\|\boldsymbol{\pi} \otimes \left(\sum_{i=1}^N \pi_i\exp(t\boldsymbol{H}_i^*)\boldsymbol{w}\right)\right\|_{\boldsymbol{\pi}} = \|\boldsymbol{\pi}\|_{\boldsymbol{\pi}}\left\|\sum_{i=1}^N \pi_i\exp(t\boldsymbol{H}_i^*)\boldsymbol{w}\right\|_2$$

$$\leq \|\boldsymbol{\pi}\|_{\boldsymbol{\pi}}\left\|\sum_{i=1}^N \pi_i\exp(t\boldsymbol{H}_i^*)\right\|_2\|\boldsymbol{w}\|_2 = \left\|\sum_{i=1}^N \pi_i\exp(t\boldsymbol{H}_i^*)\right\|_2\left\|\boldsymbol{z}^{\|}\right\|_{\boldsymbol{\pi}}$$

$$\leq (\exp(t\ell) - t\ell)\left\|\boldsymbol{z}^{\|}\right\|_{\boldsymbol{\pi}},$$

where step one follows by Part 1 of Remark 5 and step two follows by Equation 9.

**Proof of Lemma 5, Part 2** For $\forall \boldsymbol{z} \in \mathbb{C}^{Nd^2}$, we can write it as block matrix such that:

$$\boldsymbol{z} = \begin{bmatrix}\boldsymbol{z}_1 \\ \vdots \\ \boldsymbol{z}_N\end{bmatrix} = \begin{bmatrix}\boldsymbol{z}_1 \\ \vdots \\ 0\end{bmatrix} + \cdots + \begin{bmatrix}0 \\ \vdots \\ \boldsymbol{z}_N\end{bmatrix} = \sum_{i=1}^N \boldsymbol{e}_i \otimes \boldsymbol{z}_i,$$

where each $\boldsymbol{z}_i \in \mathbb{C}^{d^2}$. Please note that above decomposition is pairwise orthogonal. Applying Pythagorean theorem gives $\|\boldsymbol{z}\|_{\boldsymbol{\pi}}^2 = \sum_{i=1}^N \|\boldsymbol{e}_i \otimes \boldsymbol{z}_i\|_{\boldsymbol{\pi}}^2 = \sum_{i=1}^N \|\boldsymbol{e}_i\|_{\boldsymbol{\pi}}^2\|\boldsymbol{z}_i\|_2^2$. Similarly, we can decompose $(\boldsymbol{E}^* - \boldsymbol{I}_{Nd^2})\boldsymbol{z}$ such that

$$(\boldsymbol{E}^* - \boldsymbol{I}_{Nd^2})\boldsymbol{z} = \begin{bmatrix}\exp(t\boldsymbol{H}_1^*) - \boldsymbol{I}_{d^2} & & \\ & \ddots & \\ & & \exp(t\boldsymbol{H}_N^*) - \boldsymbol{I}_{d^2}\end{bmatrix}\begin{bmatrix}\boldsymbol{z}_1 \\ \vdots \\ \boldsymbol{z}_N\end{bmatrix} = \begin{bmatrix}(\exp(t\boldsymbol{H}_1^*) - \boldsymbol{I}_{d^2})\boldsymbol{z}_1 \\ \vdots \\ (\exp(t\boldsymbol{H}_N^*) - \boldsymbol{I}_{d^2})\boldsymbol{z}_N\end{bmatrix}$$

$$= \begin{bmatrix}(\exp(t\boldsymbol{H}_1^*) - \boldsymbol{I}_{d^2})\boldsymbol{z}_1 \\ \vdots \\ 0\end{bmatrix} + \cdots + \begin{bmatrix}0 \\ \vdots \\ (\exp(t\boldsymbol{H}_N^*) - \boldsymbol{I}_{d^2})\boldsymbol{z}_N\end{bmatrix}$$

(10)

$$= \sum_{i=1}^N \boldsymbol{e}_i \otimes \left((\exp(t\boldsymbol{H}_i^*) - \boldsymbol{I}_{d^2})\boldsymbol{z}_i\right).$$

Note that above decomposition is pairwise orthogonal, too. Applying Pythagorean theorem gives

$$\|(\boldsymbol{E}^* - \boldsymbol{I}_{Nd^2})\boldsymbol{z}\|_{\boldsymbol{\pi}}^2 = \sum_{i=1}^N \|\boldsymbol{e}_i \otimes ((\exp(t\boldsymbol{H}_i^*) - \boldsymbol{I}_{d^2})\boldsymbol{z}_i)\|_{\boldsymbol{\pi}}^2 = \sum_{i=1}^N \|\boldsymbol{e}_i\|_{\boldsymbol{\pi}}^2\|(\exp(t\boldsymbol{H}_i^*) - \boldsymbol{I}_{d^2})\boldsymbol{z}_i\|_2^2$$

$$\leq \sum_{i=1}^N \|\boldsymbol{e}_i\|_{\boldsymbol{\pi}}^2\|\exp(t\boldsymbol{H}_i^*) - \boldsymbol{I}_{d^2}\|_2^2\|\boldsymbol{z}_i\|_2^2 \leq \max_{i\in[N]}\|\exp(t\boldsymbol{H}_i^*) - \boldsymbol{I}_{d^2}\|_2^2\sum_{i=1}^N \|\boldsymbol{e}_i\|_{\boldsymbol{\pi}}^2\|\boldsymbol{z}_i\|_2^2$$

$$= \max_{i\in[N]}\|\exp(t\boldsymbol{H}_i^*) - \boldsymbol{I}_{d^2}\|_2^2\|\boldsymbol{z}\|_{\boldsymbol{\pi}}^2 = \max_{i\in[N]}\|\exp(t\boldsymbol{H}_i) - \boldsymbol{I}_{d^2}\|_2^2\|\boldsymbol{z}\|_{\boldsymbol{\pi}}^2,$$

which indicates

$$\|(\boldsymbol{E}^* - \boldsymbol{I}_{Nd^2})\boldsymbol{z}\|_{\boldsymbol{\pi}} = \max_{i \in [N]} \|\exp(t\boldsymbol{H}_i) - \boldsymbol{I}_{d^2}\|_2 \|\boldsymbol{z}\|_{\boldsymbol{\pi}} = \max_{i \in [N]} \left\| \sum_{j=1}^{+\infty} \frac{t^j \boldsymbol{H}_i^j}{j!} \right\|_2 \|\boldsymbol{z}\|_{\boldsymbol{\pi}}$$

$$\leq \left( \sum_{j=1}^{+\infty} \frac{t^j \ell^j}{j!} \right) \|\boldsymbol{z}\|_{\boldsymbol{\pi}} = (\exp(t\ell) - 1) \|\boldsymbol{z}\|_{\boldsymbol{\pi}}.$$

Now we can formally prove Part 2 of Lemma 5 by:

$$\left\| \left( \left( \boldsymbol{z}^{\|*} \boldsymbol{E} \widetilde{\boldsymbol{P}} \right)^* \right)^{\perp} \right\|_{\boldsymbol{\pi}} = \left\| \left( \left( \boldsymbol{E}^* \boldsymbol{z}^{\|} \right)^{\perp *} \widetilde{\boldsymbol{P}} \right)^* \right\|_{\boldsymbol{\pi}} \leq \lambda \left\| \left( \boldsymbol{E}^* \boldsymbol{z}^{\|} \right)^{\perp} \right\|_{\boldsymbol{\pi}} = \lambda \left\| \left( \boldsymbol{E}^* \boldsymbol{z}^{\|} - \boldsymbol{z}^{\|} + \boldsymbol{z}^{\|} \right)^{\perp} \right\|_{\boldsymbol{\pi}}$$

$$= \lambda \left\| \left( (\boldsymbol{E}^* - \boldsymbol{I}_{Nd^2}) \boldsymbol{z}^{\|} \right)^{\perp} \right\|_{\boldsymbol{\pi}} \leq \lambda \left\| (\boldsymbol{E}^* - \boldsymbol{I}_{Nd^2}) \boldsymbol{z}^{\|} \right\|_{\boldsymbol{\pi}} \leq \lambda (\exp(t\ell) - 1) \left\| \boldsymbol{z}^{\|} \right\|_{\boldsymbol{\pi}}.$$

The first step follows by Part 2 of Remark 5, the second step follows by Part 1 on Lemma 4 and the forth step is due to $\left( \boldsymbol{z}^{\|} \right)^{\perp} = \boldsymbol{0}$.

**Proof of Lemma 5, Part 3** Note that

$$\left\| \left( \left( \boldsymbol{z}^{\perp *} \boldsymbol{E} \widetilde{\boldsymbol{P}} \right)^* \right)^{\|} \right\|_{\boldsymbol{\pi}} = \left\| \left( \boldsymbol{E}^* \boldsymbol{z}^{\perp} \right)^{\|} \right\|_{\boldsymbol{\pi}} = \left\| \left( \boldsymbol{E}^* \boldsymbol{z}^{\perp} - \boldsymbol{z}^{\perp} + \boldsymbol{z}^{\perp} \right)^{\|} \right\|_{\boldsymbol{\pi}} = \left\| \left( (\boldsymbol{E}^* - \boldsymbol{I}_{Nd^2}) \boldsymbol{z}^{\perp} \right)^{\|} \right\|_{\boldsymbol{\pi}}$$

$$\leq \left\| (\boldsymbol{E}^* - \boldsymbol{I}_{Nd^2}) \boldsymbol{z}^{\perp} \right\|_{\boldsymbol{\pi}} \leq (\exp(t\ell) - 1) \left\| \boldsymbol{z}^{\perp} \right\|_{\boldsymbol{\pi}},$$

where the first step follows by Part 1 of Remark 5, the third step follows by $\left( \boldsymbol{z}^{\perp} \right)^{\|} = \boldsymbol{0}$, and the last step follows by Part 2 of Lemma 4.

**Proof of Lemma 5, Part 4** Simiar to Equation 10, for $\forall \boldsymbol{z} \in \mathbb{C}^{Nd^2}$, we can decompose $\boldsymbol{E}^* \boldsymbol{z}$ as $\boldsymbol{E}^* \boldsymbol{z} = \sum_{i=1}^{N} \boldsymbol{e}_i \otimes (\exp(t\boldsymbol{H}_i^*)\boldsymbol{z}_i)$. This decomposition is pairwise orthogonal, too. Applying Pythagorean theorem gives

$$\|\boldsymbol{E}^* \boldsymbol{z}\|_{\boldsymbol{\pi}}^2 = \sum_{i=1}^{N} \|\boldsymbol{e}_i \otimes (\exp(t\boldsymbol{H}_i^*)\boldsymbol{z}_i)\|_{\boldsymbol{\pi}}^2 = \sum_{i=1}^{N} \|\boldsymbol{e}_i\|_{\boldsymbol{\pi}}^2 \|\exp(t\boldsymbol{H}_i^*)\boldsymbol{z}_i\|_2^2 \leq \sum_{i=1}^{N} \|\boldsymbol{e}_i\|_{\boldsymbol{\pi}}^2 \|\exp(t\boldsymbol{H}_i^*)\|_2^2 \|\boldsymbol{z}_i\|_2^2$$

$$\leq \max_{i \in [N]} \|\exp(t\boldsymbol{H}_i^*)\|_2^2 \sum_{i=1}^{N} \|\boldsymbol{e}_i\|_{\boldsymbol{\pi}}^2 \|\boldsymbol{z}_i\|_2^2 \leq \max_{i \in [N]} \exp\left(\|t\boldsymbol{H}_i^*\|_2\right)^2 \|\boldsymbol{z}\|_{\boldsymbol{\pi}}^2 \leq \exp(t\ell)^2 \|\boldsymbol{z}\|_{\boldsymbol{\pi}}^2$$

which indicates $\|\boldsymbol{E}^* \boldsymbol{z}\|_{\boldsymbol{\pi}} \leq \exp(t\ell) \|\boldsymbol{z}\|_{\boldsymbol{\pi}}$. Now we can prove Part 4 of Lemma 5: Note that

$$\left\| \left( \left( \boldsymbol{z}^{\perp *} \boldsymbol{E} \widetilde{\boldsymbol{P}} \right)^* \right)^{\perp} \right\|_{\boldsymbol{\pi}} = \left\| \left( \left( \boldsymbol{E}^* \boldsymbol{z}^{\perp} \right)^{\perp *} \widetilde{\boldsymbol{P}} \right)^* \right\|_{\boldsymbol{\pi}} \leq \lambda \left\| \left( \boldsymbol{E}^* \boldsymbol{z}^{\perp} \right)^{\perp} \right\|_{\boldsymbol{\pi}} \leq \lambda \left\| \boldsymbol{E}^* \boldsymbol{z}^{\perp} \right\|_{\boldsymbol{\pi}} \leq \lambda \exp(t\ell) \left\| \boldsymbol{z}^{\perp} \right\|_{\boldsymbol{\pi}}.$$

$\square$

**Recursive Analysis** We now use Lemma 5 to analyze the evolution of $\boldsymbol{z}_i^{\|}$ and $\boldsymbol{z}_i^{\perp}$. Let $\boldsymbol{H}_v \triangleq \frac{f(v)(\gamma+\mathrm{i}b)}{2} \otimes \boldsymbol{I}_{d^2} + \boldsymbol{I}_{d^2} \otimes \frac{f(v)(\gamma-\mathrm{i}b)}{2}$ in Lemma 5. We can see verify the following three facts: (1) $\exp(t\boldsymbol{H}_v) = \boldsymbol{M}_v$; (2) $\|\boldsymbol{H}_v\|_2$ is bounded (3) $\sum_{v \in [N]} \pi_v \boldsymbol{H}_v = 0$.

Firstly, easy to see that

$$\exp(t\boldsymbol{H}_v) = \exp\left( \frac{tf(v)(\gamma+\mathrm{i}b)}{2} \otimes \boldsymbol{I}_{d^2} + \boldsymbol{I}_{d^2} \otimes \frac{tf(v)(\gamma-\mathrm{i}b)}{2} \right)$$

$$= \exp\left( \frac{tf(v)(\gamma+\mathrm{i}b)}{2} \right) \otimes \exp\left( \frac{tf(v)(\gamma-\mathrm{i}b)}{2} \right) = \boldsymbol{M}_v,$$

where the first step follows by definition of $\boldsymbol{H}_i$ and the second step follows by the fact that $\exp(\boldsymbol{A} \otimes \boldsymbol{I}_d + \boldsymbol{I}_d \otimes \boldsymbol{B}) = \exp(\boldsymbol{A}) \otimes \exp(\boldsymbol{B})$, and the last step follows by Equation 7.

Secondly, we can bound $\|\boldsymbol{H}_v\|_2$ by:

$$\|\boldsymbol{H}_v\|_2 \leq \left\| \frac{f(v)(\gamma+\mathrm{i}b)}{2} \otimes \boldsymbol{I}_{d^2} \right\|_2 + \left\| \boldsymbol{I}_{d^2} \otimes \frac{f(v)(\gamma-\mathrm{i}b)}{2} \right\|_2$$

$$= \left\| \frac{f(v)(\gamma+\mathrm{i}b)}{2} \right\|_2 \|\boldsymbol{I}_{d^2}\|_2 + \|\boldsymbol{I}_{d^2}\|_2 \left\| \frac{f(v)(\gamma-\mathrm{i}b)}{2} \right\|_2 \leq \sqrt{\gamma^2 + b^2},$$

where the first step follows by triangle inequality, the second step follows by the fact that $\|\boldsymbol{A} \otimes \boldsymbol{B}\|_2 = \|\boldsymbol{A}\|_2 \|\boldsymbol{B}\|_2$, the third step follows by $\|\boldsymbol{I}_d\|_2 = 1$ and $\|f(v)\|_2 \leq 1$. We set $\ell = \sqrt{\gamma^2 + b^2}$ to satisfy the assumption in Lemma 5 that $\|\boldsymbol{H}_v\|_2 \leq \ell$. According to the conditions in Lemma 1, we know that $t\ell \leq 1$ and $t\ell \leq \frac{1-\lambda}{4\lambda}$.

Finally, we show that $\sum_{v \in [N]} \pi_v \boldsymbol{H}_v = 0$, because

$$
\sum_{v \in [N]} \pi_v \boldsymbol{H}_v = \sum_{v \in [N]} \left( \frac{f(v)(\gamma + \mathrm{i}b)}{2} \otimes \boldsymbol{I}_{d^2} + \boldsymbol{I}_{d^2} \otimes \frac{f(v)(\gamma - \mathrm{i}b)}{2} \right)
$$

$$
= \frac{\gamma + \mathrm{i}b}{2} \left( \sum_{v \in [N]} \pi_v f(v) \right) \otimes \boldsymbol{I}_d + \frac{\gamma - \mathrm{i}b}{2} \boldsymbol{I}_d \otimes \left( \sum_{v \in [N]} \pi_v f(v) \right) = 0,
$$

where the last step follows by $\sum_{v \in [N]} \pi_v f(v) = 0$.

**Claim 4.** $\left\| z_i^\perp \right\|_{\boldsymbol{\pi}} \leq \frac{\alpha_2}{1-\alpha_4} \max_{0 \leq j < i} \left\| z_j^\| \right\|_{\boldsymbol{\pi}}$.

*Proof.* Using Part 2 and Part 4 of Lemma 5, we have

$$
\left\| z_i^\perp \right\|_{\boldsymbol{\pi}} = \left\| \left( \left( z_{i-1}^* \boldsymbol{E}\widetilde{\boldsymbol{P}} \right)^* \right)^\perp \right\|_{\boldsymbol{\pi}}
$$

$$
\leq \left\| \left( \left( z_{i-1}^{\|*} \boldsymbol{E}\widetilde{\boldsymbol{P}} \right)^* \right)^\perp \right\|_{\boldsymbol{\pi}} + \left\| \left( \left( z_{i-1}^{\perp*} \boldsymbol{E}\widetilde{\boldsymbol{P}} \right)^* \right)^\perp \right\|_{\boldsymbol{\pi}}
$$

$$
\leq \alpha_2 \left\| z_{i-1}^\| \right\|_{\boldsymbol{\pi}} + \alpha_4 \left\| z_{i-1}^\perp \right\|_{\boldsymbol{\pi}}
$$

$$
\leq \left( \alpha_2 + \alpha_2 \alpha_4 + \alpha_2 \alpha_4^2 + \cdots \right) \max_{0 \leq j < i} \left\| z_j^\| \right\|_{\boldsymbol{\pi}}
$$

$$
\leq \frac{\alpha_2}{1 - \alpha_4} \max_{0 \leq j < i} \left\| z_j^\| \right\|_{\boldsymbol{\pi}}
$$

$\square$

**Claim 5.** $\left\| z_i^\| \right\|_{\boldsymbol{\pi}} \leq \left( \alpha_1 + \frac{\alpha_2 \alpha_3}{1-\alpha_4} \right) \max_{0 \leq j < i} \left\| z_j^\| \right\|_{\boldsymbol{\pi}}$.

*Proof.* Using Part 1 and Part 3 of Lemma 5 as well as Claim 4, we have

$$
\left\| z_i^\| \right\|_{\boldsymbol{\pi}} = \left\| \left( \left( z_{i-1}^* \boldsymbol{E}\widetilde{\boldsymbol{P}} \right)^* \right)^\| \right\|_{\boldsymbol{\pi}}
$$

$$
\leq \left\| \left( \left( z_{i-1}^{\|*} \boldsymbol{E}\widetilde{\boldsymbol{P}} \right)^* \right)^\| \right\|_{\boldsymbol{\pi}} + \left\| \left( \left( z_{i-1}^{\perp*} \boldsymbol{E}\widetilde{\boldsymbol{P}} \right)^* \right)^\| \right\|_{\boldsymbol{\pi}}
$$

$$
\leq \alpha_1 \left\| z_{i-1}^\| \right\|_{\boldsymbol{\pi}} + \alpha_3 \left\| z_{i-1}^\perp \right\|_{\boldsymbol{\pi}}
$$

$$
\leq \alpha_1 \left\| z_{i-1}^\| \right\|_{\boldsymbol{\pi}} + \alpha_3 \frac{\alpha_2}{1 - \alpha_4} \max_{0 \leq j < i-1} \left\| z_j^\| \right\|_{\boldsymbol{\pi}}
$$

$$
\leq \left( \alpha_1 + \frac{\alpha_2 \alpha_3}{1 - \alpha_4} \right) \max_{0 \leq j < i} \left\| z_j^\| \right\|_{\boldsymbol{\pi}}.
$$

$\square$

Combining Claim 4 and Claim 5 gives

$$
\left\| z_k^\| \right\|_{\boldsymbol{\pi}} \leq \left( \alpha_1 + \frac{\alpha_2 \alpha_3}{1 - \alpha_4} \right) \max_{0 \leq j < k} \left\| z_j^\| \right\|_{\boldsymbol{\pi}}
$$

$$
(\text{because } \alpha_1 + \alpha_2 \alpha_3 / (1 - \alpha_4) \geq \alpha_1 \geq 1 ) \leq \left( \alpha_1 + \frac{\alpha_2 \alpha_3}{1 - \alpha_4} \right)^k \left\| z_0^\| \right\|_{\boldsymbol{\pi}}
$$

$$
= \|\phi\|_{\boldsymbol{\pi}} \sqrt{d} \left( \alpha_1 + \frac{\alpha_2 \alpha_3}{1 - \alpha_4} \right)^k,
$$

which implies

$$\langle \boldsymbol{\pi} \otimes \mathrm{vec}(\boldsymbol{I}_d), \boldsymbol{z}_k \rangle_{\boldsymbol{\pi}} \leq \|\boldsymbol{\phi}\|_{\boldsymbol{\pi}} \, d \left( \alpha_1 + \frac{\alpha_2 \alpha_3}{1 - \alpha_4} \right)^k.$$

Finally, we bound $\left( \alpha_1 + \frac{\alpha_2 \alpha_3}{1 - \alpha_4} \right)^k$. The same as [10], we can bound $\alpha_1, \alpha_2 \alpha_3, \alpha_4$ by:

$$\alpha_1 = \exp(t\ell) - t\ell \leq 1 + t^2 \ell^2 = 1 + t^2 (\gamma^2 + b^2),$$

and

$$\alpha_2 \alpha_3 = \lambda (\exp(t\ell) - 1)^2 \leq \lambda (2t\ell)^2 = 4\lambda t^2 (\gamma^2 + b^2)$$

where the second step is because $\exp(x) \leq 1 + 2x, \forall x \in [0, 1]$ and $t\ell < 1$,

$$\alpha_4 = \lambda \exp(t\ell) \leq \lambda(1 + 2t\ell) \leq \frac{1}{2} + \frac{1}{2}\lambda$$

where the second step is because $t\ell < 1$, and the third step follows by $t\ell \leq \frac{1-\lambda}{4\lambda}$.

Overall, we have

$$\left( \alpha_1 + \frac{\alpha_2 \alpha_3}{1 - \alpha_4} \right)^k \leq \left( 1 + t^2(\gamma^2 + b^2) + \frac{4\lambda t^2 (\gamma^2 + b^2)}{\frac{1}{2} - \frac{1}{2}\lambda} \right)^k$$

$$\leq \exp \left( kt^2 (\gamma^2 + b^2) \left( 1 + \frac{8}{1 - \lambda} \right) \right).$$

This completes our proof of Lemma 1. $\qquad\qquad\square$

## B.4 Proof of Theorem 1

**Theorem 1** (Markov Chain Matrix Chernoff Bound). *Let $\boldsymbol{P}$ be a regular Markov chain with state space $[N]$, stationary distribution $\boldsymbol{\pi}$ and spectral expansion $\lambda$. Let $f : [N] \to \mathbb{C}^{d \times d}$ be a function such that (1) $\forall v \in [N]$, $f(v)$ is Hermitian and $\|f(v)\|_2 \leq 1$; (2) $\sum_{v \in [N]} \pi_v f(v) = 0$. Let $(v_1, \cdots, v_k)$ denote a $k$-step random walk on $\boldsymbol{P}$ starting from a distribution $\phi$. Given $\epsilon \in (0, 1)$,*

$$\mathbb{P}\left[ \lambda_{\max} \left( \frac{1}{k} \sum_{j=1}^{k} f(v_j) \right) \geq \epsilon \right] \leq 4 \|\boldsymbol{\phi}\|_{\boldsymbol{\pi}} \, d^2 \exp\left( -(\epsilon^2 (1 - \lambda) k / 72) \right)$$

$$\mathbb{P}\left[ \lambda_{\min} \left( \frac{1}{k} \sum_{j=1}^{k} f(v_j) \right) \leq -\epsilon \right] \leq 4 \|\boldsymbol{\phi}\|_{\boldsymbol{\pi}} \, d^2 \exp\left( -(\epsilon^2 (1 - \lambda) k / 72) \right).$$

*Proof.* (of Theorem 1) Our strategy is to adopt complexification technique [8]. For any $d \times d$ complex Hermitian matrix $\boldsymbol{X}$, we may write $\boldsymbol{X} = \boldsymbol{Y} + \mathrm{i}\boldsymbol{Z}$, where $\boldsymbol{Y}$ and $\mathrm{i}\boldsymbol{Z}$ are the real and imaginary parts of $\boldsymbol{X}$, respectively. Moreover, the Hermitian property of $\boldsymbol{X}$ (i.e., $\boldsymbol{X}^* = \boldsymbol{X}$) implies that (1) $\boldsymbol{Y}$ is real and symmetric (i.e., $\boldsymbol{Y}^\top = \boldsymbol{Y}$); (2) $\boldsymbol{Z}$ is real and skew symmetric (i.e., $\boldsymbol{Z} = -\boldsymbol{Z}^\top$). The eigenvalues of $\boldsymbol{X}$ can be found via a $2d \times 2d$ real symmetric matrix $\boldsymbol{H} \triangleq \begin{bmatrix} \boldsymbol{Y} & \boldsymbol{Z} \\ -\boldsymbol{Z} & \boldsymbol{Y} \end{bmatrix}$, where the symmetry of $\boldsymbol{H}$ follows by the symmetry of $\boldsymbol{Y}$ and skew-symmetry of $\boldsymbol{Z}$. Note the fact that, if the eigenvalues (real) of $\boldsymbol{X}$ are $\lambda_1, \lambda_2, \cdots \lambda_d$, then those of $\boldsymbol{H}$ are $\lambda_1, \lambda_1, \lambda_2, \lambda_2, \cdots, \lambda_d, \lambda_d$. I.e., $\boldsymbol{X}$ and $\boldsymbol{H}$ have the same eigenvalues, but with different multiplicity.

Using the above technique, we can formally prove Theorem 1. For any complex matrix function $f : [N] \to \mathbb{C}^{d \times d}$ in Theorem 1, we can separate its real and imaginary parts by $f = f_1 + \mathrm{i}f_2$. Then we construct a real-valued matrix function $g : [N] \to \mathbb{R}^{2d \times 2d}$ s.t. $\forall v \in [N]$, $g(v) = \begin{bmatrix} f_1(v) & f_2(v) \\ -f_2(v) & f_1(v) \end{bmatrix}$. According to the complexification technique, we know that (1) $\forall v \in [N]$, $g(v)$ is real symmetric and $\|g(v)\|_2 = \|f(v)\|_2 \leq 1$; (2) $\sum_{v \in [N]} \pi_v g(v) = 0$. Then

$$\mathbb{P}\left[ \lambda_{\max} \left( \frac{1}{k} \sum_{j=1}^{k} f(v_j) \right) \geq \epsilon \right] = \mathbb{P}\left[ \lambda_{\max} \left( \frac{1}{k} \sum_{j=1}^{k} g(v_j) \right) \geq \epsilon \right] \leq 4 \|\boldsymbol{\phi}\|_{\boldsymbol{\pi}} \, d^2 \exp\left( -(\epsilon^2 (1 - \lambda) k / 72) \right),$$

where the first step follows by the fact that $\frac{1}{k} \sum_{j=1}^{k} f(v_j)$ and $\frac{1}{k} \sum_{j=1}^{k} g(v_j)$ have the same eigenvalues (with different multiplicity), and the second step follows by Theorem 3.[5] The bound on $\lambda_{\min}$ also follows similarly. $\qquad\square$

## Footnotes

[5] The additional factor 4 is because the constructed $g(v)$ has shape $2d \times 2d$.