[Reviews · NeurIPS 2020]

Review 1

Summary and Contributions: This paper develops concentration bounds for co-occurrence statistics collected from regular, steady-state Markov chains. The key technical step involves generalizing recent work by Garg et al. on the concentration of sums of matrices drawn from regular graphs to matrices drawn from any ergodic chain. The co-occurrence concentration bounds are shown to coincide with simulations for several classes of graphs.

Strengths: This is an important technical contribution that lays the groundwork for more application-focused analysis of co-occurrence statistics. As the authors point out, such statistics play a critical role in many modern machine learning algorithms, such as the action co-occurrences used in training the Tennenholtz-Mannor Starcraft II model. Although it is hard to predict how theoretical advances will be employed in future studies, these types of concentration bounds have played a critical role in the development of the theory of learning algorithms. The authors also present an intriguing set of questions for future work.

Weaknesses: Edit after Author Feedback: Thank you for pointing out that your bounds can be used to help with hyper-parameter selection in graph representation learning. Making this more clear will improve the paper. One concern is that the authors do not themselves demonstrate how to apply this concentration bound to improve the theoretical characterization of a machine algorithm. However, I do not think this is a true weakness, as this is often how progress is made in theory, i.e., one paper uses a bound to characterize an algorithm, another paper proposes an improved bound, the next paper improves the characterization using this improved bound, and so on.

Correctness: To the best of my understanding, the proofs and simulations appear correct.

Clarity: The paper is well-written and technically clear. There are only very minor typos and grammatical errors, that I'm sure can be cleaned up by the authors with another readthrough (e.g., Footnote 2, "staring state" -> "starting state")

Relation to Prior Work: The connections to prior work, especially in terms of concentration bounds for Markov chains and random matrices, are sketched out very clearly and the novel contribution are clearly delineated.

Reproducibility: Yes

Additional Feedback:


Review 2

Summary and Contributions: The paper proves a matrix version of the markov chain hoeffding inequality. It uses this inequality to bound the trajectory length needed to estimate concentration bounds for co-occurrence matrices of finite markov chains.

Strengths: The main strength is the technical contribution of a matrix markov-chain hoeffding inequality. This seems like a strong, novel contribution.

Weaknesses: The bulk of the length of the paper focuses on the "trivial" parts of the contribution: once the matrix hoeffding inequality is given, the rest of the analysis is very straightforward. The framing is also kinda strange: it's not obvious to me why the paper is about Co-occurrence Matrices of Markov chains when the much more interesting-and-powerful result of a general Matrix markov chain hoeffding bound is presented within. Co-occurrence Matrices of Markov chains seem like just one (straightforward) application of this powerful general result.

Correctness: As far as I could tell the proofs are correct.

Clarity: The paper is fairly clear, although it does get bogged down a bit in the proofs, particularly on page 6. The approach is more straightforward than the presentation makes it out to be.

Relation to Prior Work: Yes, the paper clearly distinguishes itself from prior work on matrix bounds. However, not enough attention is paid to prior work on co-occurrence matrices and bounds for them.

Reproducibility: Yes

Additional Feedback: It would be interesting (and, I think, a trivial extension) to consider the case of _hidden_ markov models, in which the underlying markov chain acts over one space a the co-occurrences of a function of that state (the observable state) are required. This will let you de-couple the size of the co-occurrence matrix from the size of the chain, and you could even probably get it to work for infinite-state-space chains (as long as the observable component is finite).


Review 3

Summary and Contributions: This paper establishes a concentration inequality in operator norm for the sample co-occurrence matrix C of a regular finite-state Markov chain. This is the matrix whose (i,j) entry counts the fraction of times states i and j co-occur within a time window of fixed size T, with a potential weighting by the difference of their occurrence times. The probability that C-E[C] exceeds eps is shown to be exponentially small in eps^2 * L, where L is the sample size (i.e. the total length of the chain). This concentration inequality is established as a corollary of a general result for the concentration of sample averages for a bounded symmetric-matrix-valued function applied to samples from an ergodic Markov chain. This general result is of independent interest. The argument is a generalization of Garg et al 2018, which established this type of result for a simple random walk on a regular graph. A central technical tool is the multi-matrix Golden-Thompson inequality which was developed in this previous work. The authors perform several simulation experiments to verify the qualitative dependence of their concentration bound on the state space size and mixing time.

Strengths: I think this is a strong submission to NeurIPS. The general matrix Chernoff bound of Theorem 2 can potentially be applied to a wide range of other problems outside of the co-occurrence matrix application described in this work, and this may be of broad interest to the NeurIPS community. While the main ideas of the proof draw upon previous work of Garg et al, the extension to Markov Chains that do not have a uniform stationary distribution is (in my view) substantial, and merits publication. The application to co-occurrence matrix estimation also has some interesting ideas, in particular how to address the vanishing of the spectral gap when passing to the Markov-Chain on the extended state space of T consecutive states.

Weaknesses: For the co-occurrence matrix application, it is unclear whether the approach of taking a union bound over the chain partitioned into tau(Q) groups, to address the issue of the vanishing spectral gap, leads to a tight result. The paper makes the simplifying assumption that the initial state is sampled from the stationary distribution, which may limit the potential applications of the main result.

Correctness: I believe that the results are correct.

Clarity: I find the paper to be very well written, both in its exposition of the problem and motivation and in its exposition of the technical proofs in the supplementary appendices. This paper was a pleasure to read.

Relation to Prior Work: Yes, I believe the relation to previous work is clearly and adequately discussed.

Reproducibility: Yes

Additional Feedback: There is no dependence on the state space size in Theorem 2, and I'm wondering if this result extends to an arbitrary (possibly continuous) state space with only cosmetic modifications to the proof. For example, can the spaces C^{Nd^2} in the proof be directly replaced by {C^{d^2}(omega): omega in Omega} for an arbitrary sample space Omega, where pi defines a probability distribution on Omega?


Review 4

Summary and Contributions: The authors provide a convergence rate for the co-occurrence matrices of Markov chains. A recent work (Qiu et al, 18) showed that the popular network embedding algorithm corresponded to implicit factorisation of a transformation of the co-occurrence matrix when the length of the random walk goes to infinity. This work provides a bound on the length of the walk required to achieve a good approximation. The authors also provide a proof for a Chernoff-like bound on the sums of matrix-valued random variables sampled from a regular Markov chain, generalising a recent result, which proved this for random walks on regular undirected graphs (Garg et al, STOC 18)

Strengths: Although I did not check the proofs in the supplementary in detail, the arguments seem sound to me. The authors provide additional empirical evidence for the convergence rate through experiments on simulated and real-world data. The result seems novel to me and the proofs are non-trivial. I am not sure how practical the result will prove to be as we will often not know the mixing time of the random walk on graphs we encounter in practice, but DeepWalk is a popular method and providing additional theoretical understanding of its performance is a useful contribution.

Weaknesses: The main limitation, as cited above, is in the practical use of the result provided here. I assume the utility of DeepWalk over NetMF is that the computational complexity is less than that of computing a SVD on the co-occurrence matrix. The authors could better demonstrate the utility of their result by repeating the BlogCatalog experiment from Qiu et al. As the mixing time of a random walk on BlogCatalog is known, the authors could demonstrate how this information could be used to select parameters for DeepWalk that gave comparable performance to NetMF(T=10) with less computation.

Correctness: The proofs and experiments are correct, as far as I can tell.

Clarity: I thought the paper was well-written and was able to understand the intuition of the proofs from the high-level sketches provided in the paper. Relating the presentation here with the work of Qiu et al would have been made easier if the author's had followed the notation from the earlier paper, or been more explicit about how the results in this paper relate to those of Qiu et al (e.g. Theorem 2.2 of Qiu et al shows that C converges to E[C] in probability).

Relation to Prior Work: The relationship with prior work is clear: the author's provide a convergence rate where previous work provided a convergence bound.

Reproducibility: Yes

Additional Feedback: I thank the authors for their response. Taking this and the other reviews into account, I'm happy to raise my score and recommend acceptance.


Review 5

Summary and Contributions: This work tackles the problem of estimating the co-occurrence matrix of a Markov chain. Two symbols are said to be co-occurent when they occur within a time window of chosen length T. The goal is to design an upper bound on the sample complexity of estimating this matrix from a single trajectory of observations. The contribution is twofold: - They derive a matrix Chernoff bound for Markov chains. - They show that the sample complexity of the task of estimating the co-occurrence matrix is upper bounded by O(tau (log n + log tau)/epsilon^2 ) where tau is the mixing time of the unknown chain, n is the number of states, and epsilon is the target precision. The authors finally illustrate the claims through experiment with synthetic and real data.

Strengths: - The results are crisp, the regularity assumption handles a very general class of chains, and the matrix Chernoff bound can probably find application elsewhere. - It is interesting to observe that the upper bound for the task of learning the co-occurrence matrix depends only logarithmically in the number of states.

Weaknesses: - This work would benefit from better citing other papers from the community, and comparing with the techniques therein. See the prior work section. - The following point deserves discussion. The Chernoff bound you design handles sums of random matrices. When working with sums and a regular Markovian process: since we know that this process is beta mixing, and that we can easily control the beta coefficient in terms of the mixing time, and since sums are easily amenable to blocking techniques followed by matrix concentration inequalities for independent matrices, the reviewer thinks it is worthwhile to explain what problem your bound could solve, that the above argument would not (see [43] for an example of the technique). Is it not possible prove the matrix Chernoff bound using this arguably simpler approach ? [43] Mixing Time Estimation in Reversible Markov Chains from a Single Sample Path D. Hsu, A. Kontorovich, D. Levin, Y. Peres, C. Szepesvári, G. Wolfer - Ann. Appl. Probab. 29 (2019), no. 4, 2439--2480. doi:10.1214/18-AAP1457, 2019

Correctness: - The reviewer quickly glanced over but cannot vouch for the proof in Appendix B.

Clarity: - Overall, the paper is well-written and makes for an interesting read. - The authors use all the terminology of 'aperiodic', 'irreducible', 'ergodic', and 'regular' Markov chains. L4: "a regular (aperiodic and irreducible)" L120: "A Markov chain is called an ergodic Markov chain if it is possible to eventually get from every state to every other state with positive probability" However, this latter definition seems to be similar to the (usual) one of irreducibility, such that either 'ergodic' or 'irreducible' is redundant here. - Lambda being qualified as a "spectral norm" is questionable. Unless the authors can justify their choice, this reviewer thinks it is better to use another term; for example in [8] already cited by the authors, lambda is referred to as the "spectral expansion".

Relation to Prior Work: - In [42], the authors investigate learning the matrix of pair-probabilities, (albeit with respect to a different norm) which leads to very many similarities with this problem when T=2. - Indeed, for example Claim 1 investigate Markovian properties of sliding windows, and proves a claim similar to that of Lemma 6.1 in [42] for the case T=2. It would be worth mentioning how your claim generalizes the former. [42] Statistical Estimation of Ergodic Markov Chain Kernel over Discrete State Space G. Wolfer, A. Kontorovich Bernoulli 2020+ https://arxiv.org/abs/1809.05014

Reproducibility: Yes

Additional Feedback: - At L290 the authors mention the future question of handling non-stationary chains. The reviewer would like to point to Section 3.3 of [44] which will provide techniques for doing so. - For an example of a chain whose multiplicative reversiblization (Fill) has null-spectral gap, the reviewer also refers the authors to the example given in Example 5.2 of [45] which is constructed from only three states instead of four, and which you may find interesting. - The mixing time is a priori unknown, therefore so is your upper bound. The scientist either has to assume an upper bound for it, or estimate it statistically. This could be further discussed. [44] Concentration inequalities for Markov chains by Marton couplings and spectral methods D Paulin - Electronic Journal of Probability, 2015 [45] Mathematical aspects of mixing times in Markov chains RR Montenegro, P Tetali - 2006

[Author Response · NeurIPS 2020]

**To Reviewer #1.** We appreciate your positive feedback and will revise our presentation accordingly. Our bounds can help hyper-parameter selection in graph representation learning (a running example about BlogCatalog can be found in our response to Reviewer #4). Prior to this work, the walk length of DeepWalk has to be selected by cross-validation.

**To Reviewer #2.** Thank you for your comments. We appreciate your views and we would like to clarify a few points.

**[The Paper Framing]** Our original intention is to analyze a graph representation learning algorithm, DeepWalk, which involves sampling random walks from a graph and counting the vertex co-occurrence matrix. Indeed, the generalized matrix Chernoff bounds are powerful and its proof is the most challenging part. So it turns out that we solved a more fundamental theory problem when studying the particular application. We are open to reframing the work as "Matrix Chernoff Bounds for Ergodic Markov Chains and its Application to Co-occurrence Matrices".

**[Hidden Markov Models (HMM)]** For a HMM, let us denote $Y$ and $X$ to be the space of observable states and hidden states, respectively. A HMM can be model by a Markov chain $\boldsymbol{P}'$ on $Y \times X$ such that $P'(y_{t+1}, x_{t+1}|y_t, x_t) = P(y_{t+1}|x_{t+1})P(x_{t+1}|x_t)$, where $P(y_{t+1}|x_{t+1})$ is the emission probability and $P(x_{t+1}|x_t)$ is the hidden state transition probability. If the co-occurrence matrix is defined only on the observable state space $Y$, then applying a similar proof like our Theorem 1 shows that one needs a trajectory of length $O(\tau(\log|Y| + \log\tau)/\epsilon^2)$ to achieve error bound $\epsilon$, where $\tau$ is the mixing time of Markov chain $\boldsymbol{P}'$ and the space of hidden states $X$ could be large. Moreover, the mixing time of Markov chain $\boldsymbol{P}'$ is bounded by the mixing time of the Markov chain on the hidden state space (i.e., $P(x_{t+1}|x_t)$).

**To Reviewer #3.** Thank you for your comments. We appreciate your views and we would like to clarify a few points.

**[The Tightness of the Bound]** The bound on co-occurrence matrices may not be tight. In our proof, we need to partition the chain $\boldsymbol{Q}$ into $\tau(\boldsymbol{Q})$ groups and then combine them with union bound, which probably gives a loose bound. As we have mentioned in 'Conclusion and Future Work' section (Sec. 6), it would be interesting to shave off the leading factor $\tau$ in the bound, as the mixing time $\tau$ could be large for some Markov chains.

**[Regarding Initial Distribution]** Thanks a lot for pointing out this! In our latest version, we have allowed the Markov chain to start from an arbitrary initial distribution $\phi$ rather than the stationary distribution $\boldsymbol{\pi}$. And there will be an additional term measuring the distance between $\phi$ and $\boldsymbol{\pi}$ in our new bound.

**[Markov Chains with Continuous States]** For infinite or continuous Markov chain, it appears to us this is a non-trivial extension and is thus beyond the scope of our paper (but certainly very interesting direction for future study). Technically speaking proving such results requires a non-trivial extension of the matrix bound (Theorem 3 in our paper), and this requires a lot more work and not the main purpose of current paper.

**To Reviewer #4.** Thank you for your comments. We appreciate your views and we would like to clarify a few points.

**[Regarding Mixing Time]** We agree that the mixing time of Markov chains are usually unknown in advance. However, it can be estimated statistically, e.g., [41]. Empirically, many real-world networks have the rapid mixing property.

**[The BlogCatalog Experiment from Qiu et al.]** Our bounds on trajectory length $L$ in Theorem 1 (with explicit constant) is $L \geq 576(\tau + T)(3\log n + \log(\tau + T))/\epsilon^2 + T$. The error bound $\epsilon$ might be chosen in the range of $[0.1, 0.01]$, which corresponds to $L$ in the range of $[8.4 \times 10^7, 8.4 \times 10^9]$. To verify that is a meaningful range for tuning $L$, we enumerate trajectory length $L$ from $\{10^4, \cdots, 10^{10}\}$, estimate the co-occurrence matrix with the single trajectory sampled from BlogCatalog, convert the co-occurrence matrix to the one required by NetMF, and factorize it with SVD. For node classification task, the micro-F1 when training ratio is 50% is

| Length $L$ | $10^4$ | $10^5$ | $10^6$ | $10^7$ | $10^8$ | $10^9$ | $10^{10}$ | NetMF |
|---|---|---|---|---|---|---|---|---|
| Micro-F1 (%) | 15.21 | 18.31 | 26.99 | 33.85 | 39.12 | 41.28 | 41.58 | **41.82** |

. As we can see, it is reasonable to choose $L$ in the predicted range. Due to page limit of author responses, we have to put more detailed results in our next version.

**To Reviewer #5.** Thank you for pointing us to relevant literature and techniques that we were not aware of before: our starting point was the random walk based graph embedding methods, and it's great to know that there are many more techniques that can be used to analyze them.

**[Prior Work]** Claim 1 in our paper (of how the $T$-step walk is itself a Markov chain) is indeed a generalization of Lemma 6.1 from [42], which discusses the special case of $T = 1$. We will cite and discuss all the related papers you mentioned ([42-45]), as well as how the bounds formally relate, in our next version.

**[Blocking Techniques]** Thank you for pointing us to this paper (Hsu et al. [43]) and the blocking techniques based on dividing the walk into nearly independent blocks of lenth around mixing time. We agree that this technique can also be used to analyze the convergence of co-occurrence matrices and much more: Garg et al. [11] was just one way to set up the analysis, and we will more clearly indicate this after adding the appropriate comparisons and references.

**[Regarding Initial Distribution]** This question is also raised by Reviewer #3. As we mentioned above, we have allowed the Markov chain to start from an arbitrary initial distribution in our latest version.

[Meta-Review · NeurIPS 2020]

This work tackles the problem of estimating the co-occurrence matrix of a Markov chain. The referees were unanimous in the assessment that this is a solid contribution, worthy of being accepted. The only reservations were some missing references in related work, which the authors agreed to discuss in the revision.